# LinPrim: Linear Primitives for Differentiable Volumetric Rendering

**Nicolas von Lützow**
Technical University of Munich
nicolas.von-luetzow@tum.de

**Matthias Nießner**
Technical University of Munich
niessner@tum.de

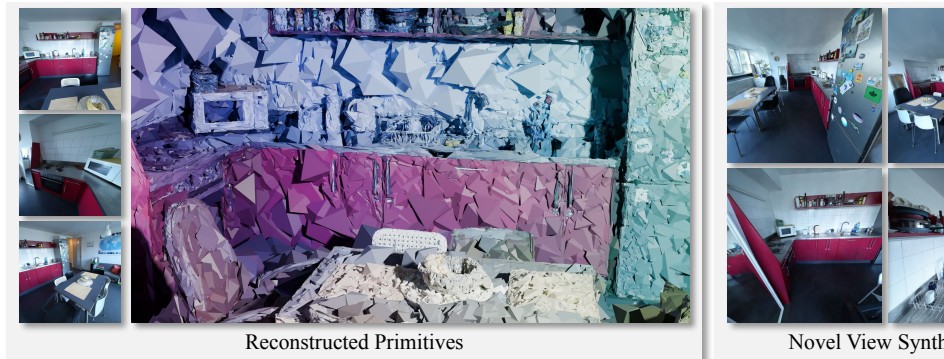

Figure 1: We present LinPrim, a new take on novel view synthesis, that leverages linear primitives - octahedra and tetrahedra - for differentiable volumetric rendering in order to facilitate 3D scene reconstruction. To this end, we propose a differentiable rendering pipeline in conjunction with a real-time capable rasterizer tailored to such primitives, thus achieving interactive frame rates for novel-view rendering and showcasing the potential of polyhedral primitives in NVS workflows.

## Abstract

Volumetric rendering has become central to modern novel view synthesis methods, which use differentiable rendering to optimize 3D scene representations directly from observed views. While many recent works build on NeRF [18] or 3D Gaussians [13], we explore an alternative volumetric scene representation. More specifically, we introduce two new scene representations based on *linear* primitives—octahedra and tetrahedra—both of which define homogeneous volumes bounded by triangular faces. To optimize these primitives, we present a differentiable rasterizer that runs efficiently on GPUs, allowing end-to-end gradient-based optimization while maintaining real-time rendering capabilities. Through experiments on real-world datasets, we demonstrate comparable performance to state-of-the-art volumetric methods while requiring fewer primitives to achieve similar reconstruction fidelity. Our findings deepen the understanding of 3D representations by providing insights into the fidelity and performance characteristics of transparent polyhedra and suggest that adopting novel primitives can expand the available design space. [1]

---

[1]Website: https://nicolasvonluetzow.github.io/LinPrim/

39th Conference on Neural Information Processing Systems (NeurIPS 2025).

# 1 Introduction

*Novel View Synthesis* (NVS) enables a holistic understanding of 3D scenes given only 2D observations. This capability is essential to a range of applications, including VR/AR, robotics, and autonomous driving. The success of state-of-the-art NVS approaches depends heavily on the chosen scene representations and the processes used to render them. In addition to visual quality, the representations have inherent characteristics that largely impact the ability to work with the reconstructed scenes in downstream applications. Explicit representations have proven particularly desirable due to their easily interpretable and manipulable structures.

Approaches in NVS utilize a wide range of scene representations that differ significantly from the conventional triangle-mesh-based pipelines used in most 3D applications. While triangle meshes are central to standard 3D content creation, their direct optimization in NVS tasks remains challenging [21, 6]. Instead, it is essential to explore the extensive design space of possible representations to assess how they impact fidelity, performance, and stability in NVS settings.

The strongest visual quality in NVS applications is currently achieved by approaches relying on *Neural Radiance Fields* (NeRF) [18, 3, 1, 2]. Although the results are impressive, NeRFs encode an implicit function, making them suboptimal for downstream applications. For instance, editing or scene animation require elements to be easily identified, accessed, or manipulated, which is inherently difficult with NeRF representations [31, 22]. Additionally, the corresponding volumetric rendering process is computationally expensive, often making real-time applications infeasible. More recently, *3D Gaussian Splatting* (3DGS) [13] introduced a real-time capable, explicit representation that also achieves strong visual performance. The 3DGS representation consists of 3D Gaussian kernels, which are optimized on known views to adjust their position and features. As evidenced by the large body of derivative works relying on 3DGS [30, 12, 17, 5, 25], the explicit nature of the representation simplifies integration into other pipelines and settings.

Motivated by the scientific curiosity of how simple, bounded primitives can be used to reconstruct high-fidelity real-world scenes, we explore an alternative to 3DGS by employing transparent polyhedra as our scene representation. Specifically, we introduce two novel scene representations based on octahedron and tetrahedron primitives. Each primitive is characterized by a compact set of features that describe its position, vertices, opacity, and view-dependent appearance. These primitives are bounded by triangular faces and of homogeneous density, which makes them an intuitive representation that can be easily understood and modified.

We build on the 3DGS pipeline to accommodate the new primitives. Our method first constructs polyhedra from a concise set of features in a fully differentiable manner. During rendering, we then rely on simple ray-triangle intersections to determine the opacity of each primitive and successively blend them. As a result, we can backpropagate image-space errors through the intersection calculations to distribute gradients onto the geometric features of each polyhedron. The optimization remains analogous to 3DGS, yet the bounded, triangular geometry of our primitives expands the applicability of differentiable rendering pipelines. Finally, we analyze the photometric reconstruction quality and performance of our approaches on real-world scenes. We are able to show comparable results while retaining real-time rendering speeds.

To sum up, our contributions are as follows:

- We introduce two novel scene representations based on transparent octahedron and tetrahedron primitives.
- We derive gradient-based optimization processes for the representations to adjust the primitives and population based on known views.
- We demonstrate that our differentiable, GPU-based rendering pipeline is real-time capable and produces high-fidelity scene reconstructions on real-world datasets.

# 2 Related Works

**Volumetric Rendering of Radiance Fields**   A major innovation in NVS has been the development of volumetric rendering approaches that represent scenes as continuous radiance fields. The seminal Neural Radiance Fields [18] demonstrated how to learn a 5D function, mapping 3D coordinates

and viewing directions to color and density, using a neural network. The rendering process used is differentiable, enabling the optimization of the representation given a known set of input views. Since its introduction, many subsequent works have improved on the initial approach. In particular, *Mip-NeRF* [1, 2] and *Zip-NeRF* [3] have drastically improved the visual quality by introducing multi-scale representations, anti-aliasing, and learned compression techniques. *Instant-NGP* [20] takes a different approach by accelerating both training and inference times via a multiresolution hash-based encoding strategy, allowing near real-time reconstruction and rendering.

Despite these advancements, NeRF-based methods remain primarily implicit, often resulting in comparatively high rendering and optimization costs. Their implicit nature can limit usability in downstream tasks. Since geometry and appearance are learned as continuous fields, no explicit surface representation is available by default. Using NeRF-based representations in downstream applications thus relies on specialized approaches or expensive post-processing [27, 26].

**Differentiable Point-based Rendering**   Point-based rendering has traditionally been seen as an alternative to mesh-based pipelines due to its simplicity in handling geometry and varying levels of surface detail [32, 10]. More recently, differentiable point-based methods have emerged, enabling end-to-end optimization of both geometry and appearance from image supervision [23, 28, 16].

The seminal work of 3D Gaussian Splatting [13] replaces traditional point primitives with elliptical Gaussian kernels. By optimizing the position and shape of the kernels from known views, they can create an explicit 3D reconstruction from known views. Additionally, each Gaussian can be efficiently "splatted" onto the screen space, allowing for real-time rendering performance. Building on the foundation of 3D Gaussian Splatting, *Mip-Splatting* [30] tackles aliasing by controlling the maximum frequency of the splats, effectively reducing flickering and dilation artifacts. This is achieved by filtering and scaling Gaussian primitives to better match the level of detail required at different viewing distances, preventing excessively sharp or overlapping splats.

Other works replace the Gaussian distributions with other primitives to achieve improved performance. *Beyond Gaussians* [4], replaces Gaussian kernels with linear kernels to achieve sharper and more precise results in high-frequency regions. Compared to our work, however, the primitives remain elliptical and non-homogeneous in density. In contrast, *EVER* [17] uses homogeneous density ellipsoids to replace alpha-blending with an exact rendering process, which avoids popping artifacts. *3D Convex Splatting* [11] employs 3D smooth convexes, completely detaching it from the Gaussian context. Their primitives offer improved flexibility and reconstruction quality.

Although these methods offer notable improvements, we instead aim to explore novel representations that rely on simple distinct primitives. While all of the above methods introduce novel primitives or rendering processes, they share the general optimization process using known views and differentiable rendering. As such, we aim to foster innovation by exploring foundational primitives to improve the general understanding of the scene representations used in NVS settings. For this, our work focuses on transparent polyhedra bounded by triangular faces, keeping compatibility with conventional geometry-processing tools while maintaining high fidelity and rendering speed.

## 3   Method

### 3.1   Linear Rendering Primitives

We introduce two representations based primarily on polyhedron primitives. For each primitive, a set of features that can accurately describe its shape needs to be defined. In our case, these include geometric parameters, position, opacity, and color.

**Octahedra.**   Importantly, there is a wide range of possible octahedra of which we only model a subset. We specifically ignore cases with non-triangular faces, which are difficult to handle and can lead to unpredictable numbers of vertices and edges. Moreover, to prevent degenerate cases, we limit the degrees of freedom by restricting vertex positions to lie on the coordinate axes relative to the center. To increase expressiveness, we instead optimize a rotation quaternion for each primitive, which ensures the shape can be oriented arbitrarily in space.

We enforce symmetry by choosing a single distance to describe opposing corners. Hence, each pair of opposite vertices shares the same distance from the center, effectively reducing the complexity

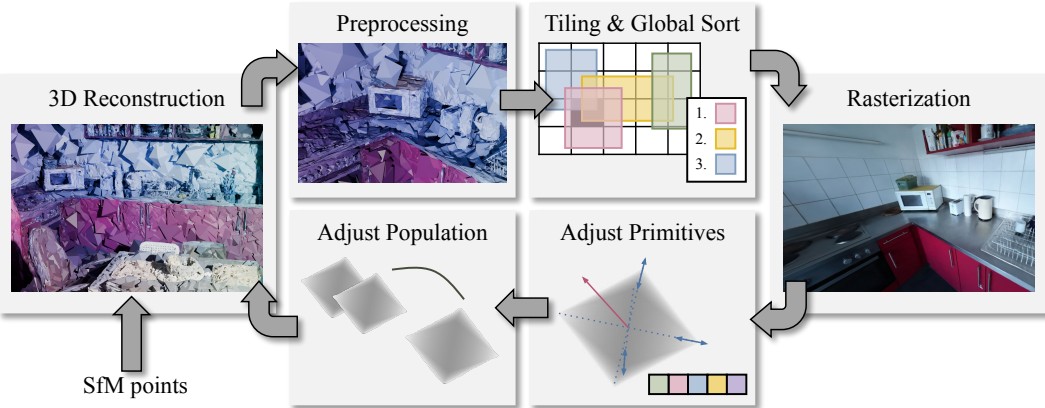

Figure 2: **Overview** of our method. Primitives are constructed from SfM points. During rendering, primitives are preprocessed according to the camera pose and sorted front-to-back. Afterward, the list is traversed and the individual color contributions are blended for each pixel. Through comparison with a known view, the primitives' features and the population are adjusted based on gradient flow.

of shape parameters. Aside from improving stability, this ensures that the center is always the true geometric center of the octahedron, simplifying downstream tasks such as bounding volume calculations and ray space approximation.

**Tetrahedra.** Although we focus primarily on octahedra, the same principles can be extended to accommodate many types of homogeneous volumes. As a particular example, we demonstrate the usage of tetrahedron primitives to reconstruct scenes. Contrary to octahedra, even regular tetrahedra are not symmetric along all axes and thus require adjustments to the stored features. In particular, we define four basis vectors, which describe directions from the center on which corners are located. All basis vectors are equally spaced such that they can represent regular tetrahedra. We then once again optimize for the distance between the center and each corner, keeping the process largely the same. Contrary to octahedra, this means that the optimized position can differ from the geometric center of the primitive.

**Memory Requirements.** In total, omitting color, each primitive is fully described by its center, rotation, corner distances, and opacity. For octahedra, this yields 11 floats in total, matching the parameter budget of a standard Gaussian kernel. Due to the absence of symmetry in our formulation, tetrahedra require 12 floats instead.

To describe colors and their view-dependent appearance, we utilize Spherical Harmonics (SH) coefficients, following the same approach described in 3DGS. Each primitive thus also stores a set of 48 SH coefficients, which aim to capture complex reflectance properties under varying angles of illumination and view. Because each coefficient is stored as a float, this set of SH parameters accounts for the majority of the memory used per primitive—by a significant margin compared to the geometric parameters. Thus, reducing memory consumption, in practice, hinges largely on either lowering the SH degree or decreasing the total number of primitives in the scene.

## 3.2 Rendering Process

We use a two-stage rendering process, beginning with per-primitive preprocessing and followed by per-pixel rasterization. This approach enables efficient parallelization by allowing many primitives to be processed in parallel before the final compositing stage.

**Preprocessing.** During preprocessing, we construct each primitive from the aforementioned features (*i.e.* the position, distances, and rotation). Given the camera pose, we then transform the primitives into

the camera's coordinate system and calculate the view-dependent color based on the viewing direction. Afterward, we project these primitives into the 3D affine ray space approximation introduced in EWA Splatting [32, 33]. Because small primitives are needed to achieve high-fidelity reconstructions, and because the approximation error grows with increasing distance from the center points, we found that the resulting error remains negligible while still reducing overall performance overhead (see Appendix D). Lastly, we compute each primitive's projected bounding box in screen space to later derive the pixels from which it is visible.

**Global Sorting and Tiling.** Following preprocessing, we leverage the tiling and global sorting techniques from 3DGS to map primitives to their corresponding pixels efficiently. We tile the screen into regions and sort primitives within each tile according to approximate front-to-back order.

**Rasterization.** During rasterization, each pixel ray either intersects a given primitive twice or not at all, since the introduced primitives are convex. To find intersections, we use the *Möller-Trumbore intersection algorithm* (MTIA) [19], which is fully differentiable and thus usable in our gradient-based optimization. If there are no intersections, the primitive is not visible; if there are two, the opacity depends on the distance between the intersection points. As our reconstruction is scale ambiguous, we must normalize the obtained intersection distances to ensure consistent appearance across different scene sizes. We do this by determining the smallest distance $d$ of the primitives and then scaling all distance measurements accordingly in a manner similar to EVER [17]. *I.e.* given the opacity of a primitive $\alpha$, we define the density of an octahedron as:

$$\sigma(\alpha) = -\frac{\log(1 - 0.99 \cdot \alpha)}{2 \cdot \min(d_x, d_y, d_z)}. \tag{1}$$

We did experiment with optimizing for the unnormalized densities directly instead of the intermediate value of the opacity, but found no performance gains in practice, while decreasing the interpretability of the feature values.

**Compositing.** Finally, we aggregate the color contributions of all visible primitives via alpha blending in front-to-back order. Because we maintain a sorted list of primitives for each tile, we can stop blending once cumulative opacity reaches a predefined threshold (*i.e.* 0.999), improving performance without noticeably affecting quality. While this approach simplifies behavior when primitives overlap, the end result is a high-fidelity rendering that faithfully reproduces fine geometry while being efficient enough for real-time applications, as shown in Appendix A.

### 3.3 Anti-Aliasing

In the context of Gaussian kernels, Mip-Splatting [30] introduced two antialiasing methods for Gaussian primitives: a 3D smoothing filter and a 2D Mip filter. In conjunction, they are able to control the maximum frequency of primitives, mitigating aliasing and dilation artifacts.

We adopt their 3D smoothing strategy to limit how small primitives can become based on visibility from the training views. As a result, each primitive is visible from at least one pixel of a training view. However, applying their 2D Mip filter approach is not possible for our primitives because they rely on modifying the screen space Gaussian distribution. Instead, for each primitive, we identify vertices that lie on the bounding box used for tiling and shift them outward, parallel to the screen. This effectively enlarges the bounding box footprint and increases the minimum size at which the primitive appears. Moreover, because the fall-off within each primitive is still treated as linear, expanding the distance from the center raises the opacity along the primitive as it is evaluated over a larger extent. Conceptually, this approach acts as a coarse approximation that serves to band-limit the high frequencies that would otherwise cause visible aliasing. Although this approach is not fully mathematically equivalent to a 2D Mip filter, it captures its function and empirically mirrors its behavior. We evaluate the effectiveness of the filters in Appendix C.

### 3.4 Optimization

We employ differentiable rendering to optimize our primitives' features from known views. For this, we use the loss formulation of 3DGS [13], consisting of L1 and SSIM terms. The gradients are then optimized using ADAM [15] and backpropagated through the entire rendering pipeline, effectively distributing the updates across the primitive features.

**Rasterization Backpropagation.** As a first step, backpropagation through the blending process yields gradients with respect to each primitive's alpha and color. When these gradients are propagated further, the alpha gradient indicates whether the primitive should be more or less transparent and, consequently, whether the intersections should move closer together or farther apart along the pixel's ray. By following the resulting intersection gradients, we can adjust each primitive's center and vertices accordingly to achieve the desired changes. This corresponds to backpropagation through the MTIA, where the intersection depth is influenced by the position of each of the corners of the triangle. In turn, the positions of the corners depend on the position of the ray space center and the offsets to it. Combining the above, by aggregating over the corners and both intersections, we can propagate gradients from alpha to the geometric features. Further details are in Appendix E.

**Preprocessing Backpropagation.** By this point, the gradients describe the loss w.r.t. the features that were originally passed to the rasterization step. While they already describe desired changes to the position of the center and corners, they do so in ray space. As a consequence, propagation through the preprocessing first involves reversing the projection and view transformations, yielding world space gradients. The final center gradients are a combination of the propagated ray space center gradients and the impact of the position on the ray space approximation and view-dependent color. Finally, the gradient flow w.r.t. to the corners splits when undoing the rotation, onto quaternion and distance features, respectively. As our scenes are scale ambiguous, we use the maximum distance between two training cameras to approximate their size and adjust the learning rates for position and distances accordingly.

### 3.5 Population Control

While the above optimization process can effectively optimize a set of primitives, it cannot create or remove them if required. In addition to an initialization scheme, this raises the need for additional processes to control the population of primitives. While we generally adapt the processes introduced by 3DGS, they require adjustments to accommodate the novel primitives.

**Initialization.** We initialize a primitive for each *Structure-from-Motion* (SfM) [24] point using its position and color. Every primitive is constructed with equal distance features to all corners, using the distance between the corresponding SfM point and its closest neighbor as a starting point. While Gaussian kernels are circular when initialized and thus unaffected by rotations, our kernels are not. Thus, we initialize the primitives with a uniformly random rotation quaternion to create a more homogeneous spatial coverage.

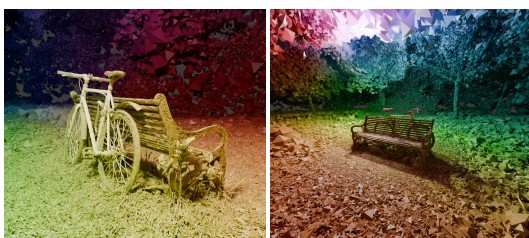

Figure 3: **Visualization** of our scene representation. We show all octahedra with an opacity of at least $0.25$ on an optimized Mip-NeRF 360 [2] Bicycle scene reconstruction.

**Adaptive Population Control.** Efficient population control requires processes to prune undesired primitives and create ones that can improve fidelity. For pruning, primitives are removed if they are too transparent, too large relative to the scene extent, or take up too much screen space of a known view. This necessitates defining the size of a primitive to determine these criteria. For octahedra, we define their size as their longest axis, *i.e.* twice the size of the longest distance. For tetrahedra, determining their largest possible depth is more involved, and we instead approximate using $\sqrt{2}$ times the distance to the furthest corner.

Creating new primitives is realized as either cloning or splitting existing ones. For this, primitives that have large view-space positional gradients are chosen, with smaller ones being cloned and larger ones being split based on the size of the primitives, as described above, and the size of the scene. Cloning a primitive creates a new primitive with the same features but without the gradient momentum, which is directly applicable to our representations.

Splitting, on the other hand, usually involves creating two new primitives with smaller sizes and placing them by sampling from the original Gaussian as a PDF. To adjust this for octahedra, since the

distances are aligned with the coordinate axes and symmetric, we sample the new positions from a PDF with standard deviations set to the corresponding distance and apply rotation afterwards. This way, primitives are more likely to spread along the longer dimensions of the octahedra. For tetrahedra, this is not trivially possible, and we instead set all standard deviations to half the largest distance. A sample result of the resulting population of octahedra can be seen in Figure 3.

**LinPrim-MCMC.** We aim to keep as many factors - apart from the primitives themselves - fixed between our approach and the Gaussian baseline. As such, our population control paradigms closely mirror the behavior of 3DGS. Nonetheless, we demonstrate the fact that LinPrim remains compatible with advancements made on Gaussian kernels and analyze the impact of population control by implementing a separate population control technique from GS-MCMC [14] for our octahedron primitives. They rethink population control as drawing samples from the distribution of the scene, which not only improves performance but also allows precise control over the final population size.

We again adopt their approach as close as possible to minimize undesired performance impacts. In this case, we essentially act as if our primitives are Gaussians with the same features. We account for the differences in magnitude between octahedron distances and Gaussian standard deviations by scaling down distances by a factor of 2.6. After this scaling, the size of the supposed Gaussians more closely resembles the primitives they imitate. A related concept was explored as *Distribution Alignment* in Beyond Gaussians [4].

## 4 Experiments

We evaluate our results on Mip-NeRF 360 [2] and ScanNet++ v2 [29]. For Mip-NeRF 360, we use all 9 available scenes, while for ScanNet++, we use only the first five NVS scenes with available test views. Mip-NeRF 360 primarily features forward-facing captures centered around a single object or region of interest, whereas ScanNet++ consists of indoor scenes with more diverse camera trajectories and test views located farther from the training coverage. We train and test on the official splits provided by each dataset and maintain consistent parameters across all evaluated scenes. All approaches are trained for 30k iterations.

Following prior work, we use Mip-NeRF 360 outdoor scenes at quarter resolution and indoor scenes at half resolution. For ScanNet++ scenes, we use the official toolkit to undistort the images and use them at full resolution. All approaches are trained, rendered, and evaluated at the same resolution.

While a large body of work has proposed improvements to 3DGS, we primarily focus on the original formulation, the modifications introduced by Mip-Splatting, and the convex primitives from *3D Convex Splatting* (3DCS) [11], as these are most closely aligned with the foundational changes introduced in our method. It is important to note that, although most evaluated approaches share an equal number of parameters per primitive, each 3D convex in 3DCS is defined by six points, resulting in a higher overall parameter count and making a direct comparison of parameter budgets less straightforward.

### 4.1 Reconstruction Quality

We measure reconstruction quality using the standard image metrics PSNR, SSIM, and LPIPS. On the ScanNet++ v2 dataset, our method consistently achieves scores close to 3DGS, Mip-Splatting, and 3DCS despite using significantly fewer primitives, as visible in Table 1. We also compare against GS-MCMC [14], which enables explicit control over the primitive population, and demonstrate that our primitives can effectively leverage the same population heuristics, achieving improved average performance at equal population counts. For Mip-NeRF 360 scenes, we see a similar trend of comparable performance but with a more compact representation, as can be seen in Table 2. These results align with other works that suggest that the number of used Gaussians can be drastically reduced without a negative impact on performance through the introduction of more advanced population control paradigms [8, 9].

Figure 4 illustrates that our method faithfully reconstructs many challenging regions while sometimes showing slightly lower overall fidelity. These outcomes stem, in part, from the bounded nature of our primitives, which enforce sharper, more "binary" decisions in underconstrained areas. In practice, this can lead to better depth estimates and crisper reconstructions in reflective or transparent

| Ground Truth | 3DGS | Mip-Splatting | Ours |
| --- | --- | --- | --- |

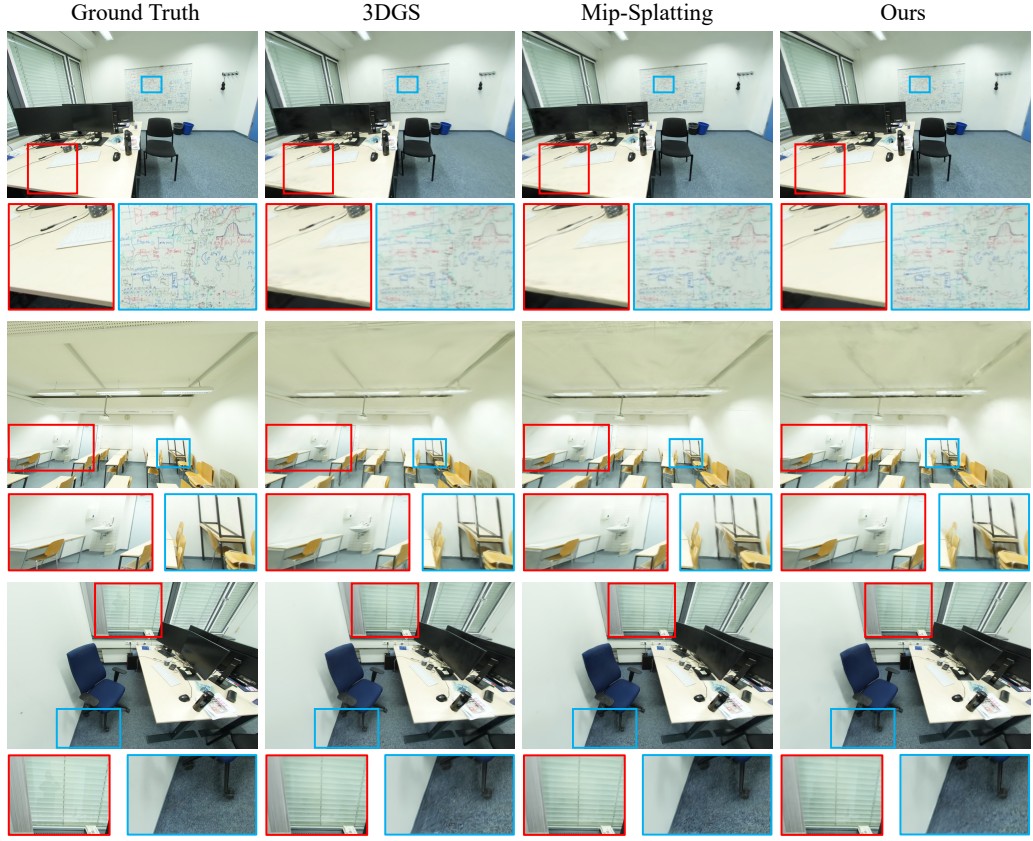

Figure 4: **Qualitative Results** on test views from ScanNet++ v2 scenes [29].

regions—such as glass or distant backgrounds—but can also result in more conspicuous edges on rarely observed surfaces like ceilings (see Appendices F and F). Overall, we find our approach to work the best in smaller, more densely captured scenes, as it retains strong geometric and visual clarity without overly densifying frequently observed regions. Further results on Mip-NeRF 360 scenes can be seen in Appendix H.

## 4.2 Tetrahedra Evaluation

In Table 3, we evaluate our tetrahedron approach using the same datasets, parameter constraints, and optimization framework described previously. By simply swapping octahedra for tetrahedra, we demonstrate that our method generalizes well to polyhedra with different geometric constraints and can still produce photorealistic reconstructions. Quantitatively, our tetrahedron approach performs similarly to octahedra on the ScanNet++ scenes and struggles on Mip-NeRF 360, highlighting the benefits of the increased symmetry and stability inherent to our octahedra. Although our existing pipeline was designed primarily around octahedra, these results confirm that it is not narrowly tailored to one specific primitive type. Nonetheless, tetrahedra could likely benefit from specialized adaptive population control and projection techniques that fully consider their asymmetric geometry, potentially closing the performance gap on currently challenging scenes. For instance, tuning how we split tetrahedra could yield further gains in accuracy.

Notably, tetrahedra and octahedra both converge to similar final primitive counts. On average, we obtained 259k tetrahedra for ScanNet++ scenes and 2.20M for Mip-NeRF 360, matching the counts of octahedra used in our experiments reasonably well. This suggests that switching between these

Table 1: **Quantitative Results** on ScanNet++ v2 scenes [29]. We limit the MCMC-based approaches to use as many primitives as LinPrim and 3DGS do on average. Our method achieves comparable performance while using much fewer primitives under default population control, and slightly higher quality when combined with MCMC-based densification. Scene identifiers are shortened for readability.

| | 39f36d | 5a269b | dc263d | 08bbbd | fb564c | Mean | Prim. |
|---|---|---|---|---|---|---|---|
| 3DGS [13] | 27.70 | 23.68 | 23.44 | 22.44 | 23.21 | 24.09 | 738k |
| Mip-Splatting [30] | 27.73 | 23.64 | 23.45 | 22.57 | 23.22 | 24.12 | 977k |
| 3DCS [11] | 27.59 | 24.19 | 22.92 | 22.78 | 23.84 | 24.26 | 440k |
| LinPrim | 27.56 | 23.84 | 22.67 | 22.31 | 23.80 | 24.04 | 255k |
| GS-MCMC [14] | 27.90 | 23.79 | 23.78 | 23.66 | 23.37 | 24.50 | 255k |
| | 27.70 | 23.40 | 23.80 | 23.44 | 22.28 | 24.12 | 738k |
| LinPrim + MCMC | 27.62 | 24.03 | 23.33 | 22.98 | 24.07 | 24.41 | 255k |
| | 27.74 | 24.13 | 23.57 | 23.33 | 23.97 | 24.55 | 738k |

Table 2: **Quantitative Results** on Mip-NeRF 360 scenes [2]. We limit the MCMC-based approaches to use as many primitives as 3DGS do on average. † 3DCS [11] uses two different sets of hyperparameters for outdoor and indoor scenes.

| | PSNR | SSIM | LPIPS | Primitives |
|---|---|---|---|---|
| 3DGS [13] | 27.43 | 0.813 | 0.217 | 3.32M |
| Mip-Splatting [30] | 27.79 | 0.827 | 0.203 | 4.17M |
| 3DCS [11] † | 27.22 | 0.801 | 0.208 | 1.02M |
| LinPrim | 26.63 | 0.803 | 0.221 | 1.79M |
| GS-MCMC [14] | 28.09 | 0.836 | 0.187 | 3.32M |
| LinPrim + MCMC | 27.04 | 0.812 | 0.211 | 3.32M |

primitive types does not inherently impose a significant memory or computational overhead. Future work on refining geometric constraints, feature parameterization, and splitting heuristics could unlock even better performance for tetrahedron-based approaches.

Table 3: **Comparison** of the performance of our Octahedron and Tetrahedron approaches on ScanNet++ v2 [29] and Mip-NeRF 360 [2] scenes. Both approaches produce comparable results on ScanNet++ scenes, but our Octahedron approach reaches higher reconstruction quality on Mip-NeRF 360 scenes.

| | ScanNet++ | | | Mip-NeRF 360 | | |
|---|---|---|---|---|---|---|
| | PSNR | SSIM | LPIPS | PSNR | SSIM | LPIPS |
| Octahedron | 24.04 | 0.849 | 0.281 | 26.63 | 0.803 | 0.221 |
| Tetrahedron | 24.05 | 0.848 | 0.302 | 25.96 | 0.790 | 0.247 |

### 4.3 Limitations

While our method demonstrates comparable results and is easily understood and modified, it is not without drawbacks. First, the bounded nature of our primitives can introduce hard edges in poorly observed regions, resulting in more "segment-like" artifacts under limited view coverage. In these regions, the ability of Gaussian kernels to smoothly transition into each other leads to more visually pleasing results, even if the reconstruction is not necessarily better. Second, while our adapted population control and MCMC-based updates modulate primitive populations well, these processes were originally designed for Gaussian primitives and are therefore not yet fully optimal for polyhedral representations. Future work could explore more specialized strategies for splitting, cloning, and sampling that better exploit the geometric structure of polyhedra. Finally, the rendering efficiency of our current implementation remains below that of the optimized Gaussian-based rasterizer (see Appendix A). Reducing the computational overhead of polygon-based rasterization by leveraging

existing optimized tools and pipelines for triangle-based rendering would substantially enhance the scalability and practicality of our approach.

**Broader Impact.** Our work provides a new approach to novel view synthesis, which extends the design space of 3D scene representations. It has the potential to improve the quality and performance of 3D reconstruction and downstream applications, benefiting research in graphics, vision, robotics, and related fields. We do not anticipate any adverse environmental, societal, or ethical effects.

# 5    Conclusion

We introduced a novel framework for volumetric scene reconstruction based on *linear primitives*, specifically octahedra and tetrahedra. Our formulation jointly optimizes geometry and appearance through a compact set of shape parameters in a fully differentiable manner. Exploiting the inherent symmetries of octahedra enables stable, high-fidelity reconstructions while maintaining a low memory footprint. In addition, our experiments with tetrahedra show that the concept generalizes naturally to other triangle-based volumes, highlighting its flexibility. Despite using significantly fewer primitives, our method achieves competitive performance across challenging datasets, demonstrating its effectiveness for real-world applications. At the same time, the bounded nature of our primitives promotes more distinct structural representations, resulting in crisper reconstructions in highly view-dependent areas and a more accurate geometric understanding, all without increasing sensitivity to initialization (see Appendices B and F).

Promising future work could focus on further leveraging the inherent properties of our primitives. For instance, their triangular faces closely resemble traditional meshes, suggesting a natural bridge between primitive-based and mesh-based representations. Developing efficient and accurate methods to binarize primitive opacity [7] could enable direct mesh optimization from images. Moreover, while the rendering enhancements introduced by EVER [17] require substantial modifications for Gaussian-based methods, our primitives naturally support intersection tests with homogeneous volumes, eliminating the need for explicit ray tracing and allowing exact blending without additional computational overhead. Leveraging existing triangle-mesh rendering pipelines could further accelerate our approach, substantially improving efficiency while removing the need for ray-space approximations (see Appendices A and D). Beyond static reconstruction, exploring downstream applications of our primitives—such as dynamic or deformable scene representations—constitutes an interesting direction for future research.

To summarize, we hope our linear primitives open up new possibilities for representing scenes in NVS applications and foster the creation of novel scene representations.

**Acknowledgements.** This work was supported by the ERC Consolidator Grant Gen3D (101171131) and the German Research Foundation (DFG) Research Unit "Learning and Simulation in Visual Computing". We thank Angela Dai for the video voice-over.

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

# A   Rendering Performance and Efficiency

**Rendering Speed.** We measure rendering speed as the average time required to render each train and test of a scene. All timings are recorded on a single RTX 3090 GPU at the same resolutions used for optimization, ensuring a fair comparison across all methods. As shown in Table 4, all evaluated approaches achieve real-time performance, demonstrating that our primitives can be rendered interactively. Our octahedron-based method reaches 68fps on ScanNet++ and 29fps on Mip-NeRF 360. Meanwhile, our tetrahedron variant achieves 175fps on ScanNet++ and 66fps on Mip-NeRF 360, making it about twice as fast as the octahedron approach due to having only half the faces to process per primitive.

**Rendering Efficiency and Trade-offs.** To further assess efficiency, we optimize variants of the same scene with different primitive counts and measure rendering speed across these populations (Table 5). While primitive subsampling would simplify comparisons, it fails to reflect realistic optimization behavior, as smaller populations are compensated by larger primitives. Rendering speed depends mainly on the number of visible primitives rather than total population: notably, GS-MCMC renders slower than 3DGS despite using roughly one third

Table 4: **Quantitative Results** on the rendering speed of tested approaches. We report the average time to render every train and test view on ScanNet++ v2 and Mip-NeRF 360 scenes.

|  | ScanNet++ | Mip-NeRF 360 |
|---|---|---|
| 3DGS [13] | 5.7ms | 8.8ms |
| Mip-Splatting [30] | 11.0ms | 20.3ms |
| LinPrim | 14.6ms | 34.6ms |
| LinPrim - Tetrahedron | 5.7ms | 15.2ms |

as many primitives. We can estimate a performance equilibrium of octahedra and Gaussians from Table 5 at around five Gaussians per octahedron, or around 1.6 faces per Gaussian, providing a starting point for future optimizations.

Table 5: **Comparison** of the rendering speed at varying primitive populations on optimized versions of ScanNet++ scene 39f36da05b. †We subsample the initial SfM points to 1/3 for the 50k and 1/2 for the 100k version since we cannot remove primitives when using MCMC densification.

|  | Primitives | Rendering Speed |
|---|---|---|
| 3DGS [13] | 795k | 4.94ms (202fps) |
| GS-MCMC [30] | 255k | 5.60ms (179fps) |
| LinPrim + MCMC | 50k † | 5.59ms (179fps) |
|  | 100k † | 8.55ms (117fps) |
|  | 250k | 15.87ms (63fps) |

**Runtime Profiling and Memory Usage.** We profile both LinPrim and 3DGS under default and MCMC-controlled populations (Table 6) to analyze runtime and memory behavior. The majority of computation time occurs during rendering, where per-tile lists are traversed to accumulate color contributions. In this stage, 3DGS only requires evaluating the function value of a Gaussian, while LinPrim calculates intersections between the pixel ray and the primitives faces. Although LinPrim processes fewer primitives per pixel, this stage is currently less optimized. Given the prevalence of ray–triangle intersection in modern graphics pipelines, integrating hardware-accelerated or existing optimized kernels offers clear potential for further speedups.

Table 6: **Performance and storage analysis** on ScanNet++ scene 39f36da05b. We report average per-frame times for preprocessing, sorting and tiling, and rendering, as well as the total number of optimized primitives and resulting file sizes.

|  | Preprocessing | Sort+Tile | Rendering | Primitives | File Size |
|---|---|---|---|---|---|
| 3DGS [13] | 0.29ms | 0.66ms | 3.99ms | 795k | 193MB |
| GS-MCMC 255k [14] | 0.18ms | 1.85ms | 3.57ms | 255k | 62MB |
| LinPrim | 0.19ms | 0.60ms | 12.02ms | 328k | 81MB |
| LinPrim + MCMC | 0.19ms | 1.08ms | 14.60ms | 250k | 62MB |
| LinPrim - Tetrahedron | 0.20ms | 0.90ms | 5.27ms | 325k | 81MB |

We evaluate the memory demands of the methods by analyzing how many primitives are kept in memory during each part of the rendering process and present the mean values on the tested views in Table 7. We also include the values when limiting either method to a similar number of total primitives (250k) by leveraging the corresponding MCMC-based approach.

As can be seen, the number of primitives in the view frustum is generally similar for both primitive types when the scene's population is similar. However, looking at the total length of the per-tile sorted lists clearly shows that the bounding boxes of LinPrim are generally much smaller, as they intersect fewer tiles. This trend is further exemplified by the total number of primitives that are iterated over. Due to the distinct boundaries of linear primitives, we can further differentiate between primitives that were only iterated over and those that were intersected and contributed to the output image. In total, linear primitives create an image with only around 5-10% of the intersections of Gaussians.

Table 7: **Analysis** of the number of primitives in memory and in the tile-list, as well as the count of total primitives iterated over and intersected by pixel rays. Output images are of size $1752 \times 1168$ with patch size $16 \times 16$, and values are evaluated on ScanNet++ v2 scene 39f36da05b.

| | Frustum | Tile-List | Iterated | Intersected |
|---|---|---|---|---|
| 3DGS [13] | 327k | 5.0M | 937M | 937M |
| GS-MCMC [14] | 113k | 5.5M | 1.3B | 1.3B |
| LinPrim | 139k | 1.4M | 275M | 52M |
| LinPrim + MCMC | 102k | 1.7M | 373M | 95M |

## B  Depth and Surface Reconstruction

**Depth Accuracy.** We evaluate geometric accuracy by comparing predicted depth maps from our method and 3DGS against ground-truth depths across all images of ScanNet++ v2 scenes. For each pixel, the predicted depth is defined as the distance to the first primitive along the viewing ray where cumulative opacity exceeds $0.5$. Ground-truth depths are derived from the official ScanNet++ meshes, and metrics are computed only at pixels with valid predictions and ground-truth values. Both methods achieve nearly complete coverage (3DGS: 99.97%, LinPrim: 99.72%) and yield visually convincing results, as shown in Figure 5. Quantitatively, LinPrim attains slightly lower L1 errors (Table 8), indicating more accurate average depth estimates, while the similar L2 errors suggest a higher presence of occasional outliers.

Table 8: **Pixel-wise depth errors** on five ScanNet++ scenes. We report per-scene L1 and L2 distances between predicted and ground-truth depth maps, computed over all valid pixels.

| | 39f36da05b | | 5a269ba6fe | | dc263dfbf0 | | 08bbbdcc3d | | fb564c935d | |
|---|---|---|---|---|---|---|---|---|---|---|
| | L1 | L2 | L1 | L2 | L1 | L2 | L1 | L2 | L1 | L2 |
| 3DGS | 0.155 | 0.268 | 0.135 | 0.209 | 0.185 | 0.313 | 0.186 | 0.288 | 0.125 | 0.215 |
| LinPrim | 0.142 | 0.263 | 0.114 | 0.202 | 0.166 | 0.318 | 0.154 | 0.285 | 0.121 | 0.238 |

**Surface Reconstruction.** To further evaluate the 3D consistency of the scene geometries, we reconstruct meshes from predicted depth maps using TSDF-Fusion with a 5 cm voxel size, followed by Marching Cubes. We then compute the Chamfer Distance to ground-truth meshes by uniformly sampling 5,000 surface points within the GT mesh boundaries. Results are summarized in Table 9. Both approaches produce coherent surfaces, with LinPrim obtaining closer matches to the GT on average.

Table 9: **Surface reconstruction accuracy** measured via Chamfer distance between meshes reconstructed from predicted depth maps on five ScanNet++ scenes.

| | 39f36da05b | 5a269ba6fe | dc263dfbf0 | 08bbbdcc3d | fb564c935d | Average |
|---|---|---|---|---|---|---|
| 3DGS | 0.126 | 0.092 | 0.096 | 0.088 | 0.093 | 0.099 |
| LinPrim | 0.070 | 0.052 | 0.123 | 0.092 | 0.060 | 0.079 |

| Ground Truth | 3DGS | Ours |

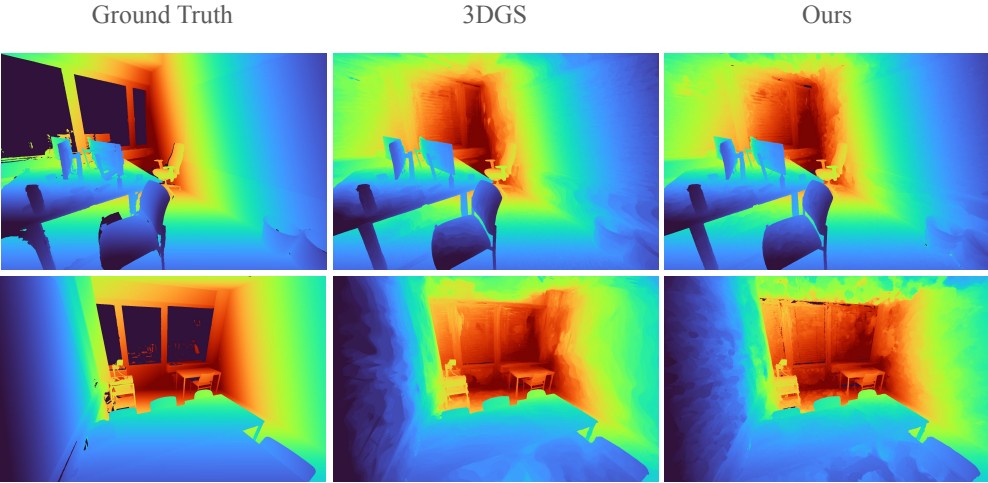

Figure 5: **Qualitative depth comparison** on ScanNet++ v2 scenes. Predicted depth maps from LinPrim and 3DGS are shown alongside depths rendered from ground truth meshes.

## C    Anti-Aliasing

**Filter Ablation.** We analyze the effect of our proposed anti-aliasing filters on reconstruction quality for representative scenes from Mip-NeRF 360 and ScanNet++ v2 (Table 10). Enabling either filter individually improves quantitative performance, most notably in PSNR, while LPIPS remains largely unchanged—indicating that perceptual structure is preserved even as pixel-level similarity improves. Combining both filters yields the strongest overall gains, confirming their complementary nature. Importantly, neither filter significantly alters the number of primitives, which indicates that these techniques refine visual fidelity without complicating the underlying geometry or increasing memory overhead.

Table 10: **Ablation** showing the impact of the two filters used in our anti-aliasing efforts. Combining both filters reduces the reconstruction error.

| Filters | | Bonsai | | | dc263dfbf0 | | |
|---|---|---|---|---|---|---|---|
| 2D | 3D | PSNR | SSIM | LPIPS | PSNR | SSIM | LPIPS |
| | | 30.84 | 0.929 | 0.196 | 22.46 | 0.870 | 0.267 |
| ✓ | | 30.96 | 0.931 | 0.196 | 22.59 | 0.870 | 0.266 |
| | ✓ | 30.97 | 0.930 | 0.195 | 22.66 | 0.871 | 0.266 |
| ✓ | ✓ | 31.21 | 0.936 | 0.195 | 22.67 | 0.871 | 0.266 |

**Zoom-Out Evaluation.** To assess robustness under scale changes, we evaluate single-scale training at progressively smaller rendering scales, following the Mip-Splatting evaluation [30].

As shown in Table 11, anti-aliasing consistently improves performance across all zoom-out factors, even though absolute gains are modest. These results show that our filtering strategy generalizes beyond the training scale and maintains visual quality in downsampled views.

## D    Ray Space Approximation

We analyze the impact of the ray-space approximation [32, 33], which our method employs to simplify the rasterization. We use MCMC-based densification [14] to minimize differences in performance caused by population control, and since the gradient magnitudes differ, which would require re-tuning hyperparameters. Furthermore, we conduct the experiments at the smallest primitive count (255k) we observed in our evaluations, as this should maximize average primitive size, and consequently the influence of the ray-space approximation, which is most accurate for small primitives.

Table 11: **Ablation** of zoom-out performance on NeRF synthetic scenes. Results are PSNR values on test set images. Approaches are optimized at full resolution and evaluated on smaller resolutions simulating zoom-outs.

| Chair | Full Res. | 1/2 Res. | 1/4 Res. | 1/8 Res. | Avg. |
|---|---|---|---|---|---|
| No AA | 35.67 | 34.39 | 30.29 | 27.17 | 31.88 |
| LinPrim | 35.70 | 34.47 | 30.42 | 27.28 | 31.97 |

| Drums | Full Res. | 1/2 Res. | 1/4 Res. | 1/8 Res. | Avg. |
|---|---|---|---|---|---|
| No AA | 25.96 | 26.42 | 25.34 | 23.14 | 25.22 |
| LinPrim | 25.98 | 26.44 | 25.43 | 23.31 | 25.29 |

| Ficus | Full Res. | 1/2 Res. | 1/4 Res. | 1/8 Res. | Avg. |
|---|---|---|---|---|---|
| No AA | 34.49 | 34.38 | 31.02 | 26.17 | 31.52 |
| LinPrim | 34.53 | 34.39 | 31.14 | 26.45 | 31.63 |

As shown in Table 12, disabling the approximation yields minor increases in reconstruction quality. However, our simple implementation incurs considerable runtime costs from removing the approximation: on the ScanNet++ scene 39f36da05b, the average rendering time increases from 15.87ms (with ray space) to 29.15ms (without ray space). This increase comes solely from the rasterization step, which increases from 14.60ms to 28.34ms. A more efficient implementation of the rendering could likely achieve the improved quality without the drastic impact on performance by leveraging efficient hardware-accelerated or optimized kernels.

Table 12: **Ablation** of the influence of the ray space approximation [32, 33] on ScanNet++ v2 scenes [29]. Since the version without the ray space does not use our 2D filter, we also give results without it. Scene identifiers are shortened for readability.

| | 39f36d | 5a269b | dc263d | 08bbbd | fb564c | PSNR | SSIM | LPIPS |
|---|---|---|---|---|---|---|---|---|
| LinPrim + MCMC | 27.62 | 24.03 | 23.33 | 22.98 | 24.07 | 24.41 | 0.858 | 0.272 |
| no 2D Filter | 27.60 | 24.01 | 23.52 | 22.95 | 23.99 | 24.42 | 0.857 | 0.272 |
| no Ray Space | 27.62 | 24.14 | 23.64 | 23.01 | 24.07 | 24.50 | 0.857 | 0.272 |

# E   Gradient Computation

In the following we provide the gradient computation of the intersection depths $\mathbf{i}$ w.r.t. to the ray space corners $\mathbf{v}^{\text{ray}}$ that span a triangle. After passing through the blending process, gradients are propagated onto the intersection depths following the opacity calculation:

$$o(\mathbf{i}_1, \mathbf{i}_2) = 1 - \exp(-\sigma(\alpha) \cdot (\mathbf{i}_2 - \mathbf{i}_1)). \tag{2}$$

From here, we derive the impact on either intersection and aggregate afterward. Given the barycentric coordinates $\{1 - u - v, u, v\}$ of the intersection, the resulting partial derivatives are:

$$\frac{\partial \mathbf{i}}{\partial \mathbf{v}_0^{\text{ray}}} = \begin{pmatrix} -\left(\frac{\partial u}{\partial \mathbf{v}_{0,x}^{\text{ray}}} + \frac{\partial v}{\partial \mathbf{v}_{0,x}^{\text{ray}}}\right) \cdot \mathbf{v}_{0,z}^{\text{ray}} + \frac{\partial u}{\partial \mathbf{v}_{0,x}^{\text{ray}}} \cdot \mathbf{v}_{1,z}^{\text{ray}} + \frac{\partial v}{\partial \mathbf{v}_{0,x}^{\text{ray}}} \cdot \mathbf{v}_{2,z}^{\text{ray}} \\ -\left(\frac{\partial u}{\partial \mathbf{v}_{0,y}^{\text{ray}}} + \frac{\partial v}{\partial \mathbf{v}_{0,y}^{\text{ray}}}\right) \cdot \mathbf{v}_{0,z}^{\text{ray}} + \frac{\partial u}{\partial \mathbf{v}_{0,y}^{\text{ray}}} \cdot \mathbf{v}_{1,z}^{\text{ray}} + \frac{\partial v}{\partial \mathbf{v}_{0,y}^{\text{ray}}} \cdot \mathbf{v}_{2,z}^{\text{ray}} \\ 1 - u - v \end{pmatrix}, \tag{3}$$

$$\frac{\partial \mathbf{i}}{\partial \mathbf{v}_1^{\text{ray}}} = \begin{pmatrix} -\left(\frac{\partial u}{\partial \mathbf{v}_{1,x}^{\text{ray}}} + \frac{\partial v}{\partial \mathbf{v}_{1,x}^{\text{ray}}}\right) \cdot \mathbf{v}_{0,z}^{\text{ray}} + \frac{\partial u}{\partial \mathbf{v}_{1,x}^{\text{ray}}} \cdot \mathbf{v}_{1,z}^{\text{ray}} + \frac{\partial v}{\partial \mathbf{v}_{1,x}^{\text{ray}}} \cdot \mathbf{v}_{2,z}^{\text{ray}} \\ -\left(\frac{\partial u}{\partial \mathbf{v}_{1,y}^{\text{ray}}} + \frac{\partial v}{\partial \mathbf{v}_{1,y}^{\text{ray}}}\right) \cdot \mathbf{v}_{0,z}^{\text{ray}} + \frac{\partial u}{\partial \mathbf{v}_{1,y}^{\text{ray}}} \cdot \mathbf{v}_{1,z}^{\text{ray}} + \frac{\partial v}{\partial \mathbf{v}_{1,y}^{\text{ray}}} \cdot \mathbf{v}_{2,z}^{\text{ray}} \\ u \end{pmatrix}, \tag{4}$$

$$\frac{\partial \mathbf{i}}{\partial \mathbf{v}_2^{\text{ray}}} = \begin{pmatrix} -\left(\frac{\partial u}{\partial \mathbf{v}_{2,x}^{\text{ray}}} + \frac{\partial v}{\partial \mathbf{v}_{2,x}^{\text{ray}}}\right) \cdot \mathbf{v}_{0,z}^{\text{ray}} + \frac{\partial u}{\partial \mathbf{v}_{2,x}^{\text{ray}}} \cdot \mathbf{v}_{1,z}^{\text{ray}} + \frac{\partial v}{\partial \mathbf{v}_{2,x}^{\text{ray}}} \cdot \mathbf{v}_{2,z}^{\text{ray}} \\ -\left(\frac{\partial u}{\partial \mathbf{v}_{2,y}^{\text{ray}}} + \frac{\partial v}{\partial \mathbf{v}_{2,y}^{\text{ray}}}\right) \cdot \mathbf{v}_{0,z}^{\text{ray}} + \frac{\partial u}{\partial \mathbf{v}_{2,y}^{\text{ray}}} \cdot \mathbf{v}_{1,z}^{\text{ray}} + \frac{\partial v}{\partial \mathbf{v}_{2,y}^{\text{ray}}} \cdot \mathbf{v}_{2,z}^{\text{ray}} \\ v \end{pmatrix}. \tag{5}$$

In the case of octahedra, since the vertices are either in positive or negative direction from the center, the gradient onto the actual ray space feature can be negated, which we omit for the sake of readability.

To obtain the derivatives of the barycentric coordinates w.r.t. to the triangle corners, we backpropagate through the MTIA [19]. Below, we give the results for corner $\mathbf{v}_0$, the formulas for $\mathbf{v}_1$ and $\mathbf{v}_2$ are analogous. Notably, the barycentric coordinates are invariant to the depth and as such have only two non-zero components. Let $\mathbf{r}$ denote pixel ray coordinates and $d$ the determinant, then:

$$\frac{\partial u}{\partial \mathbf{v}_0} = \frac{1}{d}\left(\begin{pmatrix}\mathbf{v}_{2,y} - \mathbf{r}_y \\ \mathbf{r}_x - \mathbf{v}_{2,x}\end{pmatrix} - u\begin{pmatrix}\mathbf{v}_{2,y} - \mathbf{v}_{1,y} \\ \mathbf{v}_{1,x} - \mathbf{v}_{2,x}\end{pmatrix}\right), \quad \frac{\partial v}{\partial \mathbf{v}_0} = \frac{1}{d}\left(\begin{pmatrix}\mathbf{r}_y - \mathbf{v}_{1,y} \\ \mathbf{v}_{1,x} - \mathbf{r}_x\end{pmatrix} - v\begin{pmatrix}\mathbf{v}_{2,y} - \mathbf{v}_{1,y} \\ \mathbf{v}_{1,x} - \mathbf{v}_{2,x}\end{pmatrix}\right). \tag{6}$$

Gradient flow towards the ray space center follows the same process as to that of the corners. Since they are just matrix multiplications, undoing ray space and view-space transformations is straightforward. Finally, once the gradients have been mapped back into world space, we propagate each corner's gradient into the primitive's distance and rotation parameters.

## F  Robustness and Failure Analyses

**Sensitivity to Initialization.** We evaluate the robustness of LinPrim and 3DGS to initialization perturbations by modifying the structure-from-motion (SfM) input used for primitive initialization. The ablation is conducted on the first ScanNet++ scene 39f36da05b, where both methods achieve comparable baseline performance. We test three variants: (i) *Noisy Points*, where Gaussian noise with a standard deviation of 5% of the maximum camera distance (about 14 cm) is added to each SfM point; (ii) *Half Points*, where half of the SfM points are randomly removed; and (iii) *Half-Size*, where initial primitive sizes are halved, reducing spatial coverage. As shown in Table 13, all of the above methods reduce performance across all metrics and for either approach. Noticeably, reducing the coverage through smaller initial primitives hurts 3DGS's performance more than ours. This suggests that even though LinPrim boundaries are distinct and intersections are fewer, relevant visual cues can be transferred onto the primitives without issue.

Table 13: **Sensitivity to initialization.** We evaluate robustness to initialization on ScanNet++ scene 39f36da05b by comparing reconstructions from perturbed training setups.

| | 3DGS | | | LinPrim | | |
|---|---|---|---|---|---|---|
| | PSNR | SSIM | LPIPS | PSNR | SSIM | LPIPS |
| Base | 27.70 | 0.859 | 0.211 | 27.56 | 0.857 | 0.225 |
| Noisy Points | 27.61 | 0.858 | 0.217 | 27.54 | 0.855 | 0.228 |
| Half Points | 27.69 | 0.859 | 0.214 | 27.51 | 0.854 | 0.232 |
| Half-Size | 27.25 | 0.849 | 0.235 | 27.44 | 0.856 | 0.227 |

**Per-Class Reconstruction and Failure Cases.** To better understand where each approach excels or struggles, we compute per-class PSNR on ScanNet++ v2 using semantic renderings from the first three scenes. Results for a selection of classes are shown in Table 14. In absolute terms, the failure cases of our approach were the classes of *Storage Cabinet* and *Computer Tower*, in which LinPrim achieved the lowest relative performance compared to 3DGS. On the contrary, the classes that exhibit the strongest view-dependent appearance in the scenes - windows (+0.41), doors (+1.59), and monitors (+2.77) - are better reconstructed by LinPrim. There further seem to be types of geometry and appearance LinPrim is better able to capture, *e.g.*, whiteboards (+1.95), blinds (+0.64), and heaters (+0.96). These results suggest that linear and Gaussian primitives capture complementary appearance and geometry properties, and that hybrid representations combining both may alleviate such edge-case failures.

**Effect of Spherical Harmonics Degree.** We evaluate the impact of the spherical harmonics degree on performance for LinPrim and 3DGS by optimizing scenes for either approach with reduced maximal degrees and comparing test-view performance. We show results in Table 15 and omit SSIM and LPIPS results for brevity reasons, and since they are less impacted by the changes to spherical harmonics than PSNR.

Higher SH degrees do not always improve reconstruction quality and can even harm performance in less observed regions by overfitting appearance while masking geometric inaccuracies. This trend

Table 14: **Per-class reconstruction quality** on the first three ScanNet++ scenes. We report PSNR for 3DGS and LinPrim across selected semantic categories, along with relative differences and pixel counts per class.

| | 3DGS | LinPrim | Relative | Pixels |
|---|---|---|---|---|
| Wall (Top 1) | 33.74 | 34.76 | +1.02 | 2.7M |
| Floor (Top 2) | 27.80 | 25.80 | -2.00 | 1.1M |
| Table (Top 3) | 32.65 | 32.88 | +0.22 | 646k |
| Storage Cabinet | 36.50 | 32.22 | -4.28 | 283k |
| Monitor | 26.50 | 29.27 | +2.77 | 223k |
| Window | 23.36 | 23.77 | +0.41 | 110k |
| Computer Tower | 34.93 | 23.41 | -11.52 | 15k |

is more evident on ScanNet++, where test views are farther from the training distribution than on Mip-NeRF 360. In our ablation, we find that 3DGS generally benefits more from spherical harmonics than LinPrim. As seen in Table 14, LinPrim nonetheless reaches higher visual quality in more view-dependent areas. In conjunction, this can likely be attributed to a better understanding of the underlying surfaces and geometry instead of an affinity to spherical harmonics.

Table 15: **Ablation** on the impact of reduced SH-degrees on the PSNR on three ScanNet++ and two Mip-NeRF 360 scenes, and compared to the degree 3 baselines.

| | SH-degree | ScanNet++ | | | Mip-NeRF 360 | |
| | | 39f36da05b | 5a269ba6fe | dc263dfbf0 | Bicycle | Room |
|---|---|---|---|---|---|---|
| 3DGS | 2 | -0.38 | -0.55 | -0.58 | -0.64 | -0.76 |
| | 1 | -0.37 | -0.41 | -0.64 | -0.79 | -0.88 |
| | 0 | -0.29 | -0.37 | -0.41 | -1.01 | -1.00 |
| LinPrim | 2 | +0.05 | +0.21 | -0.22 | -0.18 | -0.09 |
| | 1 | +0.01 | +0.15 | -0.22 | -0.47 | -0.39 |
| | 0 | -0.01 | +0.10 | -0.33 | -0.89 | -0.58 |

## G   Experimental Setting

In this section, we give specific values for the hyperparameters used and the experimental setting to ensure the reproducibility of our results. Experiments were performed using either a single RTX A6000 or RTX 3090, rendering speed was evaluated using the same RTX 3090 setup for all approaches.

We use the following learning rates for both our Octahedron and Tetrahedron approaches, the position learning rates and the corresponding schedule are consistent with 3DGS [13]. Specific values were chosen empirically.

Table 16: The learning rates used for our Octahedron and Tetrahedron approaches. † The distance learning rate is scaled by the maximum distance between a known camera pose and the mean camera pose to ensure consistent behavior even in scale-ambiguous settings.

| | Learning Rate |
|---|---|
| Color | $2.5 \times 10^{-3}$ |
| SH Coefficients | $1.25 \times 10^{-4}$ |
| Opacity | $2.5 \times 10^{-2}$ |
| Rotation | $1 \times 10^{-3}$ |
| Distance † | $2.6^{-1} \times 10^{-4}$ |

Other notable paradigms and parameters concerning the initialization and population of primitives are as follows:

- Primitives are initialized with distances equal to the nearest SfM point, clamped between $10^{-5}$ and $0.5$. Opacities are set to $0.1$ and rotations are sampled uniformly at random.
- The population is adjusted every 250 iterations. Primitives with position gradients larger than $1.5 \times 10^{-4}$ are considered for densification, and ones with opacity smaller than $0.025$ are removed. Similarly, primitives with a size larger than $40\%$ of the scene or 20 pixels are removed.
- During densification, primitives smaller than $1\%$ of the scene are duplicated. Larger ones are replaced with two newer ones with a relative size of $1.2^{-1}$. For octahedra, the position is chosen normally at random with a standard deviation aligned with, and equal to, the distances in each dimension. For tetrahedra, we sample normally with standard deviations equal to half of the largest distance.
- We do not propagate gradients through the density normalization factor (see Equation 1).
- For our anti-aliasing efforts, we set the kernel size of the 2D filter to $0.1$. For octahedra, the ray space vector is moved by half that amount, as the adjustment is mirrored on both sides. The 3D filter remains consistent with Mip-Splatting.

## H Additional Results

In Figure 6, we show qualitative results of our representation on Mip-NeRF 360 scenes. In Table 17, we show the average PSNR, SSIM, and LPIPS metrics on the used ScanNet++ v2 scenes.

Table 17: **Quantitative Results** on ScanNet++ v2 scenes [29]. We limit the MCMC-based approaches to use as many primitives as LinPrim and 3DGS do on average. The presented results are the average over the five scenes considered during our evaluation.

|  | PSNR | SSIM | LPIPS | Primitives |
|---|---|---|---|---|
| 3DGS [13] | 24.09 | 0.853 | 0.263 | 738k |
| Mip-Splatting [30] | 24.12 | 0.852 | 0.261 | 977k |
| 3DCS [11] | 24.26 | 0.848 | 0.273 | 440k |
| LinPrim | 24.04 | 0.849 | 0.281 | 255k |
| GS-MCMC [14] | 24.50 | 0.862 | 0.256 | 255k |
|  | 24.12 | 0.861 | 0.251 | 738k |
| LinPrim + MCMC | 24.41 | 0.858 | 0.272 | 255k |
|  | 24.55 | 0.860 | 0.263 | 738k |

## I Existing Assets

- **3DGS** [13]: Code can be accessed under `https://github.com/graphdeco-inria/gaussian-splatting` and uses a custom License also found in the repository. For our experiments and code, we use commit *d9fad7b*.
- **Mip-Splatting** [30]: Code can be accessed under `https://github.com/autonomousvision/mip-splatting` and follows the 3DGS license. For our experiments and code, we use commit *dda02ab*.
- **3DGS-MCMC** [14]: Code can be accessed under `https://github.com/ubc-vision/3dgs-mcmc` and follows the 3DGS license. For our experiments and code, we use commit *7b4fc9f*.
- **Mip-NeRF 360 Data** [1]: Available from `https://jonbarron.info/mipnerf360/` and does not provide license terms.
- **ScanNet++ v2 Data** [29]: Available from `https://kaldir.vc.in.tum.de/scannetpp/` and uses a custom license also found on the website.

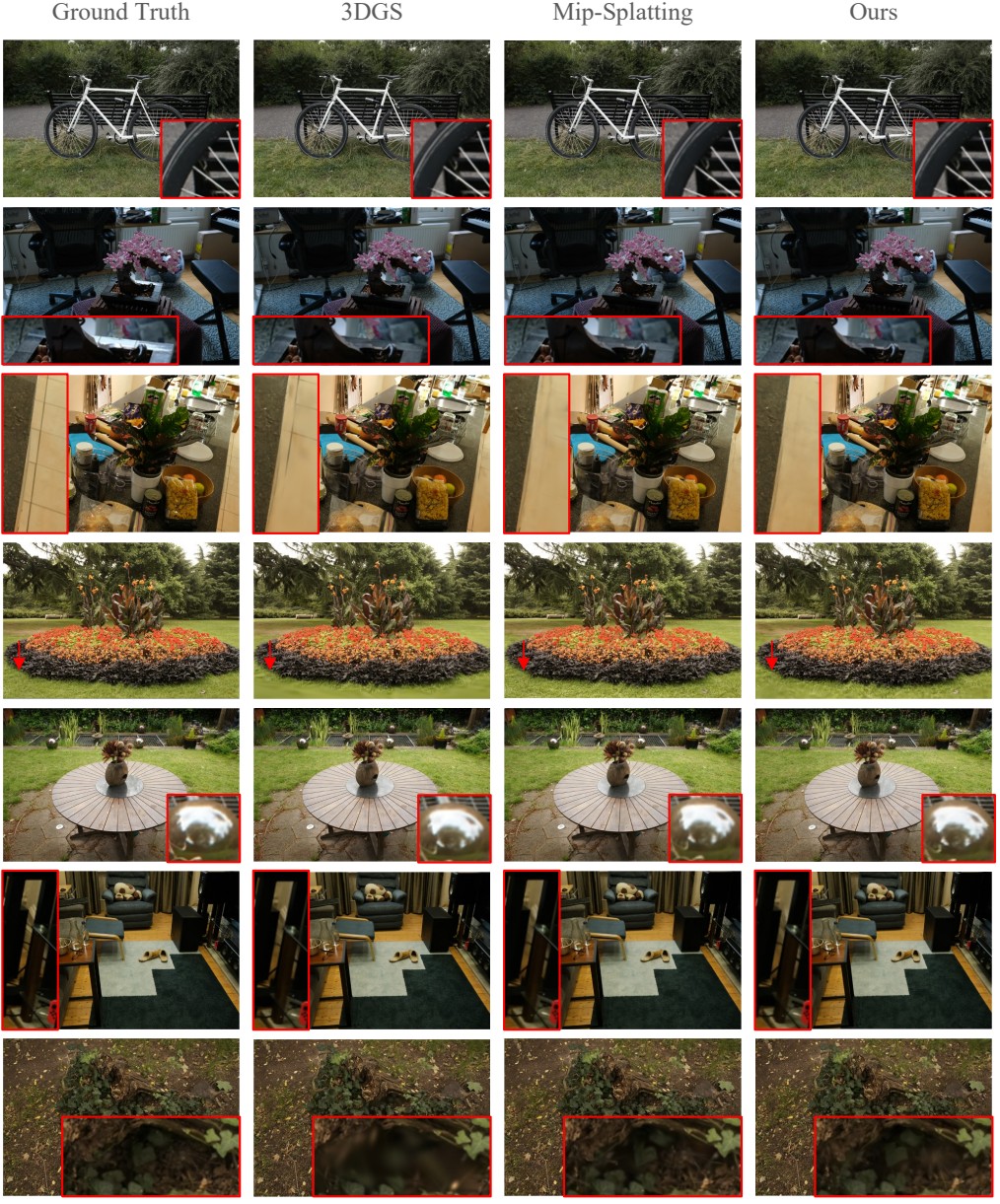

Figure 6: **Qualitative Results** on test views from Mip-NeRF 360 scenes [2].

