# OpenReview forum: "LinPrim: Linear Primitives for Differentiable Volumetric Rendering"
_NeurIPS.cc/2025/Conference — NeurIPS 2025 poster_

### Official Review · Reviewer_Z51C · 2025-06-23

**Clarity:** 1
**Significance:** 2
**Originality:** 2
**Rating:** 4
**Confidence:** 2

**Summary:**

This paper proposes two new primitives to perform volumetric rendering, namely octahedra and tetrahedra, and adapt the existing 3D Gaussian Splatting (3DGS) framework to select a suitable number of primitives and to optimize the rendering process efficiently. The method is evaluated on two datasets, showing comparable performance to competitive baselines while, in some cases, reducing the number of primitives required.

**Questions:**

- the population control technique seems to be potentially complex, what is the impact on the training time and stability of the overall approach?
- the comparison between using tetrahedra and octahedra could be improved, what are the advantages and disadvantages of one compared to the other based also on the empirical evidences?
- why are the baselines in Table 1 and 2 different? What is the performance of those not reported in one of the two tables?

**Ethical Concerns:**

["NO or VERY MINOR ethics concerns only"]

**Final Justification:**

I thank the authors for their effort in answering my questions.

I have also studied the other reviews and respective answers and realized that most of my concerns (especially related to lack of details and clarity) are definitely not shared by other reviews. This is likely due to my limited understanding of the paper and my limited background which gave me a hard time judging the work.

I am thus willing to increase my rating above the acceptance threshold.

**Limitations:**

yes

**Paper Formatting Concerns:**

no concerns

**Quality:**

2

**Strengths And Weaknesses:**

**Strengths:**
- the method is evaluated on multiple datasets and ablation studies are presented
- visualizations aid the explanation of the method

**Weaknesses:**
- the paper is overall difficult to follow: several concepts are not introduced (e.g., primitives, alpha blending, rasterization, quaternions, Structure-from-Motion) and the method section goes into detail too quickly (e.g., octahedra and tetrahedra are not introduced in Sec. 3.1, which only presents their technical challenges for the problem at hand)
- the contribution appears limited: the paper adapts the existing 3DGS framework to accommodate 2 new primitives, without significantly diverging from it. The numerical results also do not show a significant enough performance improvement to justify this as a relevant contribution (e.g., on ScanNet++, the performance and number of primitives is very similar to GS-MCMC)
- the experimental section is overall not clear: the two datasets are not introduced (what do they contain, what is the specific task) and some choices appear arbitrary:
	- why are the baselines for the two datasets different?
	- what are the metrics in Table 1?

---

> ### Author Rebuttal · Authors · 2025-07-31
>
> We thank the reviewer for the insightful comments. We address the questions and concerns below:
>
> **Population Control.** Our aim was to keep as many factors apart from the primitives themselves fixed when comparing to 3DGS to make the evaluation as fair as possible, and focus on the underlying primitive type. As such, our population control paradigms closely mirror the behavior of 3DGS, with changes necessitated by the change of primitive.
>
> To analyze the impact of the population control on performance, we implement a completely separate population control technique from the 3D-MCMC paper \[NeurIPS 2024\], which rethinks population control as drawing samples from the distribution of the scene. This not only improves performance but also allows precise control over the final population size. We once again adopt their approach as close as possible to minimize undesired performance impacts. In this case, we essentially act as if our primitives are Gaussian-like with standard deviations equal to the linear distances divided by 2.6. This value was chosen as it results in a close match in appearance between the original shape and its Gaussian substitute.
>
> As can be seen in Table 11, this approach is able to consistently improve performance when evaluating on the first three ScanNet++ scenes. We further believe that this illustrates how approaches made as improvements to 3DGS can be similarly applied to LinPrim. We will add these results to the revised manuscript.
>
> Table 11: Ablation of the performance of MCMC-based densification on linear primitives with a fixed number of 250k and 1M primitives in the scene on the first three ScanNet++ scenes.
>
> |  | 39f36da05b |  |  | 5a269ba6fe |  |  | dc263dfbf0 |  |  |
> | :---- | :---- | :---- | :---- | :---- | :---- | :---- | :---- | :---- | :---- |
> |  | SSIM | PSNR | LPIPS | SSIM | PSNR | LPIPS | SSIM | PSNR | LPIPS |
> | **LinPrim** | 0.857 | 27.56 | 0.225 | 0.825 | 23.85 | 0.310 | 0.871 | 22.67 | 0.266 |
> | **LinPrim-MCMC 250k** | 0.857 | 27.65 | 0.227 | 0.834 | 24.05 | 0.299 | 0.881 | 23.35 | 0.256 |
> | **LinPrim-MCMC 1M** | **0.862** | **27.67** | **0.212** | **0.838** | **24.13** | **0.284** | **0.883** | **23.61** | **0.247** |
>
> **Comparison between Octahedra and Tetrahedra.** We will include the results of our experiments on comparing octahedra and tetrahedra-based approaches in the final manuscript. For purposes of readability, we include only the results on three ScanNet++ and Mip-NeRF 360 scenes in Table 12\.
>
> Based on the quantitative and qualitative results, we find that octahedra and tetrahedra generally show similar visual characteristics in equally populated areas of a scene. Differences arise when considering the population control paradigms. They are required to enhance the initial coverage of the scene and enable higher fidelity in well-observed regions. Using the symmetry of the octahedra primitives allows us to mirror established approaches for Gaussian primitives to place new primitives which is not directly possible for tetrahedra.
>
> Nonetheless, we believe that tetrahedra promise a large improvement in runtime efficiency (see Table 8\) when compared to octahedra due to their reduced face count, with the current drawbacks largely concerning them being less related to established approaches. As such, applying novel or well-suited population control approaches to tetrahedra offers a promising avenue for future work.
>
> Table 12: Comparison of the performance of LinPrim’s octahedron and tetrahedron-based approaches on three ScanNet++ and three Mip-NeRF 360 scenes.
>
> |  | 39f36da05b |  |  | 5a269ba6fe |  |  | dc263dfbf0 |  |  |
> | :---- | :---- | :---- | :---- | :---- | :---- | :---- | :---- | :---- | :---- |
> |  | SSIM | PSNR | LPIPS | SSIM | PSNR | LPIPS | SSIM | PSNR | LPIPS |
> | **Octahedra** | **0.857** | **27.56** | **0.225** | 0.825 | **23.85** | **0.310** | **0.871** | 22.67 | 0.266 |
> | **Tetrahedra** | 0.853 | **27.56** | 0.250 | **0.828** | 23.71 | 0.335 | 0.871 | **22.94** | **0.286** |
>
> |  | Bicycle |  |  | Treehill |  |  | Room |  |  |
> | :---- | :---- | :---- | :---- | :---- | :---- | :---- | :---- | :---- | :---- |
> |  | SSIM | PSNR | LPIPS | SSIM | PSNR | LPIPS | SSIM | PSNR | LPIPS |
> | **Octahedra** | **0.739** | **24.76** | **0.236** | **0.617** | **21.80** | **0.346** | **0.913** | **29.88** | **0.216** |
> | **Tetrahedra** | 0.722 | 24.29 | 0.275 | 0.612 | 21.41 | 0.361 | 0.899 | 29.05 | 0.240 |
>
> **Used Baselines.** We thank the reviewer for his feedback and will update our experiments section to make it clearer and more readable. Keeping space limitations in mind, the baselines were chosen following the availability of results and relevance to our work. We will ensure that a more complete overview of results and introduction of the datasets will be available in the final manuscript.

---

> > ### Comment · Area_Chair_ir6v · 2025-08-03
> >
> > Reviewer Z51C, did the rebuttal address your concerns? Do you have any further questions or comments for the authors?

---

> > ### Comment · Reviewer_Z51C · 2025-08-04
> >
> > I thank the authors for their effort in answering my questions.
> >
> > I have also studied the other reviews and respective answers and realized that most of my concerns (especially related to lack of details and clarity) are definitely not shared by other reviews. This is likely due to my limited understanding of the paper and my limited background which gave me a hard time judging the work.
> >
> > I am thus willing to increase my rating above the acceptance threshold.

---

> > > ### Author Response · Authors · 2025-08-05
> > >
> > > Thank you, we appreciate the reconsideration and updated rating and will include the additional results in the revised paper.

---

### Official Review · Reviewer_EARy · 2025-06-26

**Clarity:** 3
**Significance:** 3
**Originality:** 3
**Rating:** 5
**Confidence:** 4

**Summary:**

This paper introduces LinPrim, a novel explicit scene representation for novel-view synthesis that replaces Gaussian kernels with transparent linear polyhedra. LinPrim achieves competitive reconstruction quality on ScanNet++ and Mip-NeRF 360, while using significantly fewer primitives than 3DGS.

**Questions:**

See the weaknesses.

**Ethical Concerns:**

["NO or VERY MINOR ethics concerns only"]

**Final Justification:**

The author provided a detailed and persuasive response. I think this article has good potential and encourage the author to apply the representation to other fields (such as 4D reconstruction, Feed-forward generation, etc.). I maintain my score to accept.

**Limitations:**

yes.

**Paper Formatting Concerns:**

no.

**Quality:**

3

**Strengths And Weaknesses:**

Strengths
1. The method provides a complete and systematic solution, including full forward and backward derivations for triangle-ray intersections, seamless integration into the 3DGS training loop, and full preservation of end-to-end differentiability.
2. This approach enables a marked decrease in attribute complexity while preserving high-fidelity rendering performance.

Weaknesses
1. Recent work like 3D Convex Splatting [CVPR 2025] also propose modifications to Gaussian primitives. A more in-depth comparison and analysis with these methods would strengthen the paper.
2. Although the method employs fewer primitives, its training and inference times are still slower than those of 3DGS. The potential reasons, including per-triangle computation costs or overdraw effects, are not thoroughly investigated.
3. Despite emphasizing a substantial reduction in primitive count, the paper omits quantitative measurements of total storage requirements, making the practical memory efficiency unclear.
4. It would be interesting to explore whether this representation provides enhanced geometric fidelity relative to 3DGS, such as in depth maps or mesh reconstruction.

---

> ### Author Rebuttal · Authors · 2025-07-31
>
> We thank the reviewer for their positive feedback and insightful questions. We address the reviewer’s questions below:
>
> **Comparison to 3DCS.** We will add a full quantitative and qualitative comparison of 3D Convex Splatting to our evaluation and discuss the approach. For the sake of brevity, we include only a partial comparison of the approaches on the first three ScanNet++ scenes in Table 7\. Keeping in mind that 3DCS was not tuned on this dataset, we are able to achieve comparable performance. Including densification paradigms via Markov Chain Monte Carlo (as further described in our comment to reviewer *Z51C*), we are able to slightly outperform their results. In the revised paper, we will extend this comparison to all ScanNet++ and Mip-NeRF 360 scenes and include additional visualizations.
>
> Table 7: Quantitative evaluation on three ScanNet++ scenes.
>
> |  | 39f36da05b |  |  | 5a269ba6fe |  |  | dc263dfbf0 |  |  |
> | :---- | :---- | :---- | :---- | :---- | :---- | :---- | :---- | :---- | :---- |
> |  | **SSIM** | **PSNR** | **LPIPS** | **SSIM** | **PSNR** | **LPIPS** | **SSIM** | **PSNR** | **LPIPS** |
> | **3DGS** | 0.859 | **27.701**  | **0.211** | 0.829 | 23.678 | 0.290 | 0.879 | 23.440 | **0.246**  |
> | **3DCS** | 0.857 | 27.624 | 0.212 | 0.828 | **24.219** | 0.290 | 0.868 | 22.766 | 0.264 |
> | **LinPrim** | 0.857 | 27.557 | 0.225 | 0.825 | 23.845 | 0.310  | 0.871  | 22.668 | 0.266 |
> | **LinPrim-MCMC** | **0.862** | 27.666 | 0.212 | **0.838** | 24.127 | **0.284** | **0.883** | **23.611** | 0.247 |
>
> **Performance.** To provide deeper insights into our method’s runtime characteristics, we profiled the evaluation of our method and 3DGS. We evaluate the methods as-is, as well as using the MCMC population control to ensure similar populations for both approaches, as can be seen in Table 8\.
>
> Our results show that the large majority of the cost is incurred in the rendering stage, where per-tile lists are traversed to aggregate color contributions. In this stage, 3DGS only requires evaluating the function value of a Gaussian, while LinPrim calculates intersections between the pixel ray and the primitives faces. As a consequence, even though LinPrim traverses fewer primitives per pixel than 3DGS (as shown in Table 6), the current implementation is simply less optimized. Since ray-triangle intersections are commonly required, we believe leveraging existing optimized implementations for this use presents a significant opportunity for runtime improvements and will discuss the findings in the revised manuscript.
>
> **Storage Requirements.** As shown in Table 8, our representation not only uses fewer primitives in general it also employs smaller bounding boxes and consequently holds far fewer primitives in memory at any given time. Concerning total storage requirements, the actual disk size of an optimized scene is almost solely reliant on the number of primitives. Looking at the file sizes of the optimized scenes, we find that given unrestricted optimizations, the LinPrim scene is around 1/3 of the 3DGS size.
>
> Table 8: Analysis of the runtime of individual steps in the rendering pipeline and corresponding primitive counts and file sizes on the ScanNet++ scene 39f36da05b.
>
> |  | Preprocessing | Sorting & Tiling | Rendering | \#Primitives | File Size |
> | :---- | :---- | :---- | :---- | :---- | :---- |
> | **3DGS** | 0.29ms | 0.66ms | 3.99ms | 795k | 193MB |
> | **GS-MCMC 255k** | **0.18ms** | 1.85ms | **3.57ms** | 255k | **62MB** |
> | **LinPrim** | 0.19ms | **0.60ms** | 12.02ms | 328k | 81MB |
> | **LinPrim-MCMC 250k** | 0.19ms | 1.08ms | 14.60ms | **250k** | **62MB** |
> | **LinPrim-Tetrahedra** | 0.20ms | 0.90ms | 5.27ms | 325k | 81MB |
>
> **Depth Maps and Mesh Reconstruction.** We evaluate the accuracy of depth maps predicted by our method and by 3DGS on all images from the first three ScanNet++ scenes. For each pixel, the predicted depth is defined as the depth of the first primitive along the ray where cumulative opacity exceeds 0.5. The resulting depth maps look convincing for both approaches.
>
> Ground‑truth depths are extracted from the ScanNet++ meshes via the official toolkit. We compute error metrics only at pixels with valid ground‑truth and prediction values \- pixels missing either are excluded. Both methods achieve similarly high coverage, with 3DGS covering 99.97 % of pixels and LinPrim covering 99.79 %. Results can be seen in Table 9\.
>
> Table 9: Pixel-wise depth map errors on three ScanNet++ scenes.
>
> |  | 39f36da05b |  | 5a269ba6fe |  | dc263dfbf0 |  |
> | :---- | :---- | :---- | :---- | :---- | :---- | :---- |
> |  | *L1* | *L2* | *L1* | *L2* | *L1* | *L2* |
> | **3DGS** | 0.1553 ± 0.1124 | 0.2678 ± 0.1519 | 0.1352 ± 0.0749 | 0.2086 ± 0.0944 | 0.1854 ± 0.0511 | **0.3126** ± 0.0905 |
> | **LinPrim** | **0.1415** ± 0.1079 | **0.2634** ± 0.1469 | **0.1143** ± 0.0790 | **0.2022** ± 0.1081 | **0.1662** ± 0.0408 | 0.3183 ± 0.0935 |
>
> To assess the 3D consistency and extract surfaces from the predicted depth maps, we apply TSDF-Fusion with a voxel size of 5cm and apply Marching Cubes to reconstruct meshes from the images. We then compute the Chamfer Distance to the ground-truth meshes from ScanNet++ by uniformly sampling 5000 surface points, as can be seen in Table 10\.
>
> Table 10: Chamfer distances between mesh meshes reconstructed on three ScanNet++ scenes by fusing predicted depth maps.
>
> | *Chamfer Distance* | 39f36da05b | 5a269ba6fe | dc263dfbf0 |
> | :---- | :---- | :---- | :---- |
> | **3DGS** | 0.1264 | 0.0924 | **0.0959** |
> | **LinPrim** | **0.0700** | **0.0518** | 0.1232 |
>
> The results indicate that LinPrim produces slightly more accurate depth estimates than 3DGS, suggesting better geometric reconstruction. The Chamfer Distance evaluation shows that LinPrim achieves better 3D reconstruction in two out of three scenes. This indicates that while depth predictions are generally of higher quality, exact results can vary based on scene characteristics. We will add qualitative and quantitative results and discuss them in our revised manuscript.

---

> > ### Comment · Reviewer_EARy · 2025-08-05
> >
> > I appreciate the author's detailed response. In addition to the quantitative comparison, for the next version, I suggest that the author could add some depth visualizations and qualitative comparisons to make the results more substantial.

---

> > > ### Author Response · Authors · 2025-08-05
> > >
> > > Thanks, we have already rendered the visualizations and are including them in the revised manuscript.

---

### Official Review · Reviewer_HtCm · 2025-06-30

**Clarity:** 2
**Significance:** 3
**Originality:** 3
**Rating:** 5
**Confidence:** 3

**Summary:**

This manuscript proposes to use linear primitives for 3D scene reconstruction and novel view synthesis. Inspired by 3D Gaussian Splatting, this work designs a differentiable rasterization pipeline and a population control strategy for the linear primitives, enabling gradient-based optimization. Compared to vanilla 3D Gaussian Splatting, the proposed method achieves comparable rendering quality with fewer primitives, while exhibiting qualitatively superior fidelity in some challenging cases, such as sharper geometric edges and mitigation of over-densification (e.g., the table edge and carpet in Fig.4).

**Questions:**

1.	The manuscript does not clearly explain how and why EWA splatting is applied on linear primitives. L149 mentions a trade-off between approximation error and computational overhead, it is better to provide a quantitative comparison.

2.	Why is opacity maintained by each primitive and use it to derive density, rather than directly learning a density attribute?

3.	Although 3DGS also have a truncated range, its AABB is somewhat over-estimated. However, the proposed linear primitive seems incapable of receiving gradients from non-intersecting pixels. Could this property impose stricter requirements on initialization?

**Ethical Concerns:**

["NO or VERY MINOR ethics concerns only"]

**Final Justification:**

Most of my concerns have been resolved by the rebuttal. Taking into account both the results presented in the submission and responses, I think the proposed primitive constitutes a non-trivial improvement over vanilla 3DGS. Therefore, I decide to raise my rating to accept.

**Limitations:**

yes

**Quality:**

3

**Strengths And Weaknesses:**

Strengths:

1.	The proposed method can achieve close rendering quality with significantly fewer primitives. In some challenging cases, as shown in the qualitative comparisons, it exhibits unique advantages over previous arts.

2.	The proposed linear primitive holds strong potential to inspire future exploration. Compared to 3DGS, it allows for more precise define of ray-primitive intersection, while easier to equipping topological structuring, and more like the triangle meshes commonly used in graphics pipelines.

Weaknesses：

1.	As shown in Table 4, the advantage of using fewer primitives does not transform into rendering efficiency (although it can already achieve interactive frame rates on standard benchmarks). In addition, since L321 claims the advantage of low memory footprint, it is suggested to compare the memory usage quantitatively to better illustrate the compactness of the representation.

2.	The benefits of the proposed linear primitives over the vanilla 3D Gaussians are not convincingly described. Now its potential is far from fully unleashed, and the demonstrated advantages appear circumstantial rather than fundamental or indispensable.

---

> ### Author Rebuttal · Authors · 2025-07-31
>
> We thank the reviewer for the precise and perceptive analysis and comments. We address the comments below:
>
> **How and why is EWA splatting applied?** We employ EWA splatting by using the ray-space approximation to cut down per-primitive work during rasterization. Although our primitives remain full 3D objects, projecting them into ray space lets us treat each triangle as an effectively 2D shape for intersection and gradient computations. Concretely, this simplifies intersection calculations as the ray directions are always orthogonal to the image plane and have only one nonzero component. This effectively means that testing a ray against a triangle reduces to a 2D problem. Additionally, this applies similarly to the gradient calculations, where position gradients collapse to two dimensions until transformed out of ray space.
>
> We agree that an ablation study comparing the rendering speed and quality, each with and without the ray-space approximation, would help clarify the trade-offs and will include the corresponding quantitative results and discussion in the final manuscript.
>
> **Optimizing Densities.** Despite the representations being largely equivalent in our use case, optimizing a density directly is certainly plausible and might help performance by avoiding unnecessary transformations. We will run an ablation on this to see if there are positive effects in practice.
>
> For the submission, we used opacities for two main reasons:
>
> 1. They are bounded, and we thus assume them to be more numerically stable during optimization.
> 2. Compared to densities, opacities are more easily interpretable, making it easier to understand distributions of primitives or contributions to final appearance.
>
> **Sensitivity to Initialization.** To evaluate the sensitivity of our approach and 3DGS to initialization, we apply changes to the initialization scheme and evaluate the relative performance impact. We perform this ablation on the first ScanNet++ scene (39f36da05b), on which both approaches performed similarly well.
>
> These are the specifics of the changes made:
>
> 1. Noisy Points: We add noise to the position of the SfM points used for initialization. For this, we sample random offsets in each dimension with a standard deviation of 5% of the maximum distance between cameras, which is around 14cm for the scene.
> 2. Half \#Points: We randomly drop half of the SfM points used for initialization.
> 3. Half-Size Primitive Initialization: The initial size of the primitives in either approach depends on the distance between neighbouring SfM points. For this test, we reduce the initial size of primitives by half, leading to sparse initial coverage of the 3D space.
>
> As can be seen in Table 5, all of the above methods reduce performance across all metrics and for either approach. Noticeably, reducing the coverage through smaller initial primitives hurts 3DGS's performance more than ours. This suggests that even though LinPrim boundaries are distinct and intersections are fewer, relevant visual cues can be transferred onto the primitives. In general, while the scope of the experiment is limited, we believe it showcases that our approach is not more dependent on a strong initialization and will include the results in the revised manuscript.
>
> Table 5: Ablation on the impact of worse initialization schemes on the final reconstruction quality on ScanNet++ scene 39f36da05b.
>
> |  | 3DGS |  |  | LinPrim |  |  |
> | :---- | :---- | :---- | :---- | :---- | :---- | :---- |
> |  | **SSIM** | **PSNR** | **LPIPS** | **SSIM** | **PSNR** | **LPIPS** |
> | **Base** | **0.859** | **27.70** | **0.211** | **0.857** | **27.56** | **0.225** |
> | **Noisy Points** | 0.858 | 27.61 | 0.217 | 0.855 | 27.54 | 0.228 |
> | **Half \#Points** | **0.859** | 27.69 | 0.214 | 0.854 | 27.51 | 0.232 |
> | **Half-Size Primitive Init.** | 0.849 | 27.25 | 0.235 | 0.856 | 27.44 | 0.227 |
>
> **Memory Usage and Bounding Boxes.** We evaluate the memory demands of the methods by analyzing how many primitives are kept in memory during each part of the rendering process. For this, we run the evaluation on the first ScanNet++ scene (39f36da05b) and present the mean values on the tested views in Table 6\. We also include the values when limiting either method to a similar number of total primitives by leveraging the corresponding MCMC-based approach as introduced in our reply to reviewer *Z51C*.
>
> As can be seen, the number of primitives in the view frustum is generally similar for both primitive types when the scene's population is similar. However, looking at the total length of the per-tile sorted lists clearly shows that the bounding boxes of LinPrim are generally much smaller, as they intersect fewer tiles.
>
> This trend is further exemplified by the total number of primitives that are iterated over. Due to the distinct boundaries of linear primitives, we can further differentiate between primitives that were only iterated over and those that were intersected and contributed to the output image. In total, linear primitives create an image with only around 5-10% of the intersections of Gaussians.
>
> As Table 5 shows that this does not translate to issues with gradient flow, it suggests that the large majority of intersections in 3DGS only contribute minor effects to the output image and similarly do not enable significant gradient flow. We believe these values present a significant opportunity for future work to optimize the rendering process and leverage the concise nature of our representation. And will include the findings and discuss opportunities in our revised manuscript.
>
> Table 6: Analysis of the number of primitives in memory and in the tile-list, as well as the count of total primitives iterated over and intersected by pixel rays. Output images are of size 1752x1168 with patch size 16x16, and values are evaluated on ScanNet++ scene 39f36da05b.
>
> |  | Frustum | Tile-List | Iterated | Intersected |
> | :---- | :---- | :---- | :---- | :---- |
> | **3DGS** | 327k | 5.0M | 937M | 937M |
> | **GS-MCMC 255k** | 113k | 5.5M | 1.3B | 1.3B |
> | **LinPrim** | 139k | **1.4M** | **275M** | **52M** |
> | **LinPrim-MCMC 250k** | **102k** | 1.7M | 373M | 95M |
>
> **Benefits of Linear Primitives over 3DGS.** We kindly refer to our reply to reviewer *EARy* about geometry and surface reconstruction and reviewer *Pgum* about semantic-wise visual quality. In summary, we show that our primitives provide improvements in geometric reconstruction and are better able to reconstruct view-dependent appearance when compared to the Gaussian baseline.

---

> > ### Comment · Reviewer_HtCm · 2025-08-03
> >
> > The author’s response clarifies several key details, and the additional results address most of my raised concerns. Specifically, the provided comparison with Gaussian primitives on more challenging initialization scheme resolves my concerns about potential initialization sensitivity may brought by different characteristic of the proposed linear primitive with vanilla 3D Gaussians. Result in Table 6 confirms that consistently fewer primitives are required at each stage of the rendering pipeline, supporting the claimed lower memory footprint. Although the benefits of the proposed primitives over the vanilla 3DGS are still not logically articulated, the additional results across multiple tasks can empirically demonstrate its advantages. Therefore, I will take active consideration to further increase my rating.

---

> > > ### Author Response · Authors · 2025-08-03
> > >
> > > Thanks for the quick follow-up. We aim to better articulate the advantages. Please let us know if this clarifies things, and we would be happy to update the paper accordingly:
> > >
> > > 1. **Bounded.** Linear primitives have finite support with exact ray-face intersections. This reduces unnecessary contributions, yielding cleaner gradients and better geometric reconstructions. Empirically, this corresponds to lower depth errors (Table 9\) and closer mesh reconstructions (Table 10\) in most scenes, as well as maintained robustness to degraded initialization (Table 5).
> > > 2. **View-dependent Appearance.** LinPrim performs better on specular/view-dependent classes (Table 3\) and degrades less when the SH degree is reduced (Table 2).
> > > 3. **Compactness.** Since the primitive faces are compact, the AABBs touch far fewer image tiles and pixels, leading to large reductions in tile-list length, total iterations, and actual intersections during rendering, even when populations are similar (Table 6). This lowers the memory footprint, improves scalability, and offers a significant opportunity for future performance improvements.
> > > 4. **Compatibility.** Linear primitives are compatible with many of the advancements made to Gaussian primitives: Leveraging MCMC-based densification not only works without issue but actually significantly increases performance (Table 11). Moreover, LinPrim is more suited to some advancements (e.g., exact blending following EVER \[arXiv:2410.01804\]), promising further disproportional performance improvements in some areas.

---

> > > > ### Comment · Reviewer_HtCm · 2025-08-04
> > > >
> > > > Thanks for your further comment. I agree that these unique properties of the proposed primitive can indeed lead to clear advantages in certain aspects, thus constituting a meaningful improvement over 3DGS. Actually, my earlier concern largely stemmed from an elevated expectation, that this new primitive would not only be well compatible with recent advancements in Gaussian primitives, but has distinctive potential beyond what vanilla 3DGS can offer. This does not detract my positive recommendation.

---

### Official Review · Reviewer_Pgum · 2025-07-03

**Clarity:** 3
**Significance:** 2
**Originality:** 2
**Rating:** 4
**Confidence:** 4

**Summary:**

The paper introduces LinPrim, a differentiable 3D representation for volumetric-rendering-based reconstruction that replaces Gaussian kernels with bounded, linear polyhedral primitives: axis-aligned octahedra and tetrahedra.

Each primitive is described by 11–12 floats for geometry + 48 SH coefficients for color and is rasterized with a custom GPU kernel that (i) projects the polyhedron into the ray space used by EWA splatting, (ii) computes differentiable ray-triangle intersections, and (iii) composites in front-to-back order (with simple anti-aliasing heuristics).

As evaluated on ScanNet++ and Mip-NeRF 360 dataset, the proposed method achieves comparable (slightly off) quality while using 4x fewer primitives compared with 3DGS.

**Questions:**

- Complexity trade-off. Given octahedra are slower but fewer than Gaussians, where is the break-even point in primitive count vs. faces for equal FPS?
- Surface extraction. Is this representation friendly for surface extraction? If so, how does accuracy and performance looks like? I also assume there is better algorithm for iso-surface extraction compared with marching cubes on this representation.
- SH bandwidth. Did you experiment with degree-1 or learned MLP colour decoders? How sensitive is quality to SH order?
- Dynamic scenes. Your rasterizer is well suited to convex, bounded volumes—does it extend to per-frame primitive motion (e.g. for dynamic NeRFs) without severe overhead?
- Edge-case failures. Besides ceilings, which scene structures (thin wires, foliage, specular objects) break most often, and can hybrid primitives (polyhedra + Gaussians) alleviate that?

**Ethical Concerns:**

["NO or VERY MINOR ethics concerns only"]

**Final Justification:**

All of my concerns have been properly addressed. Please include those additional discussions and evaluations in the revised version of the paper. I vote for acceptance.

**Limitations:**

Yes

**Quality:**

3

**Strengths And Weaknesses:**

Strengths:
- It is always interesting and worth exploring to bring polyhedra-based primitives into differentiable rendering with splatting.
- The representation is compact with efficiency improvement over 3DGS. Octahedra need only 11 float params (same budget as a Gaussian) yet deliver comparable PSNR/SSIM with one quarter the primitive count on ScanNet++.
- Fully differentiable rasterizer introduced by analytical ray-triangle gradients (derived in the appendix) enable end-to-end training without resorting to volumetric sampling.
- The paper compares octahedra vs. tetrahedra, tests anti-aliasing filters, and re-uses publicly available 3DGS/Mip-Splatting code with identical training schedules for fair comparisons.

Weaknesses:
- Runtime still heavier than 3DGS. Octahedra render at ~14.6 ms per 1 K×1 K view vs. 5.7 ms for 3DGS on the same hardware; the face count largely offsets the smaller population.
- It is always a downside for polygon-based primitives to represent hard edges when coverage is sparse, whereas Gaussians fade gracefully.
- Limited evaluation scope. It would be great and more comprehensive to have more evaluations especially on the success and failure cases of the proposed representation, which would unveil unique characteristics brought by this representation compared with 3DGS. Testing on more different scenes would be beneficial, such as large-scale scenes, indoor scenes with lots of surfaces with Manhattan assumptions, and scenes with reflections, etc.
- Anti-aliasing is heuristic. The 2D bbox inflation filter is an ad-hoc substitute for Mip-Splatting’s analytic filter and might not generalize to zoom-out scenarios.

---

> ### Author Rebuttal · Authors · 2025-07-31
>
> We thank the reviewer for the thorough and insightful review. We address the individual points and questions below:
>
> **Complexity Trade-off.** To estimate the break-even point between the rendering speed of our octahedron approach and 3DGS, we optimize several variations of the same scene with different primitive counts and track the average rendering speed in Table 1\. Please refer to our reply to reviewer *Z51C*  for more details on how primitive counts were limited.
>
> While subsampling primitives for evaluating rendering speed would allow us to more easily test rendering speeds at different primitive counts, it would not accurately reflect an optimized scene since the optimization will compensate for smaller populations with larger primitives, which in turn increase rendering times. Additionally, rendering speed is largely dependent on the number of visible primitives, not the population, as clearly demonstrated by 3DGS-MCMC taking longer to render while having only around a third the population size of 3DGS.
>
> While an exact measure for relative rendering efficiency is difficult, when optimizing with the same MCMC heuristics, the ratio seems to be rendering 5 Gaussians for every octahedron or rendering 1.6 faces for each Gaussian. We kindly refer to Table 6 and Table 8 and the accompanying comments for additional insights on rendering efficiency. We will add this discussion and evaluation to the final manuscript.
>
> Table 1: Comparison of the rendering speed of ScanNet++ scene 39f36da05b at different primitive populations. All experiments were run on an RTX 3090\. *\*We subsample the initial SfM points to 1/3 for the 50k and 1/2 for the 100k version since we cannot remove primitives when using MCMC densification.*
>
> |  | \# Primitives | Rendering Speed |
> | :---- | :---- | :---- |
> | **3DGS** | 795k | **4.94ms (202fps)** |
> | **3DGS-MCMC** | 255k | 5.60ms (179fps) |
> | **LinPrim-MCMC** | **50k\*** | 5.59ms (179fps) |
> |  | 100k\* | 8.55ms (117fps) |
> |  | 250k | 15.87ms (63fps) |
>
> **Surface Extraction.** We kindly refer to our comment to reviewer *EARy*, where we analyze the quality of depth maps and surface reconstructions. In summary, we find that our method exhibits slightly stronger geometric reconstruction than the 3DGS baseline.
>
> We agree with the reviewer that our triangle-based primitives are a more natural fit for direct iso-surface extraction, and exploring specialized mesh-extraction algorithms is a promising direction for future work. As shown by prior works (e.g., 2D Gaussian Splatting \[SIGGRAPH 2024\]), there is often a trade-off between geometric accuracy and visual fidelity in reconstructed scenes. We argue that LinPrim’s explicit triangular representation aligns more closely with the goal of producing high-quality meshes, and that incorporating additional optimization constraints \- such as opacity requirements or encouraging face-normal alignment \- could further improve surface quality, possibly without or with only little sacrifice in terms of appearance. Investigating these surface-aware regularizers may unlock both sharper geometry and maintained rendering performance.
>
> **SH Bandwidth.** We evaluate the impact of the spherical harmonics degree on performance for LinPrim and 3DGS by optimizing scenes for either approach with reduced maximal degrees and comparing test-view performance. We show results in Table 2 and omit SSIM and LPIPS results for brevity reasons, and since they are less impacted by the changes to spherical harmonics than PSNR.
>
> Noticeably, higher SH-degrees don’t necessarily improve the quality of reconstructions for unobserved regions, and can instead hurt performance by obfuscating underlying geometry and overfitting on known views. We can observe this by comparing performance on ScanNet++, where test views are more out-of-distribution, to the results on Mip-NeRF 360\.
>
> In our ablation, we find that 3DGS generally benefits more from spherical harmonics than LinPrim. As seen in Table 3, LinPrim nonetheless reaches higher visual quality in more view-dependent areas. In conjunction, this can likely be attributed to a better understanding of the underlying surfaces and geometry instead of an affinity to spherical harmonics. We will add a more expansive analysis and discussion in the revised manuscript.
>
> Table 2: Ablation on the impact of reduced SH-degrees on the PSNR on three ScanNet++ and two Mip-NeRF 360 scenes, and compared to the degree 3 baselines.
>
> | *PSNR* | SH-degree | ScanNet++ |  |  | Mip-NeRF 360 |  |
> | :---- | :---- | :---- | :---- | :---- | :---- | :---- |
> |  |  | **39f36da05b** | **5a269ba6fe** | **dc263dfbf0** | **Bicycle** | **Room** |
> | **3DGS** | **2** | \-0.38 | \-0.55 | \-0.58 | \-0.64 | \-0.76 |
> |  | **1** | \-0.37 | \-0.41 | \-0.64 | \-0.79 | \-0.88 |
> |  | **0** | \-0.29 | \-0.37 | \-0.41 | \-1.01 | \-1.00 |
> | **LinPrim** | **2** | **\+0.05** | **\+0.21** | **\-0.22** | **\-0.18** | **\-0.09** |
> |  | **1** | \+0.01 | \+0.15 | **\-0.22** | \-0.47 | \-0.39 |
> |  | **0** | \-0.01 | \+0.10 | \-0.33 | \-0.89 | \-0.58 |
>
> **Dynamic Scenes.** We expect our rasterizer to be applicable to object motion and dynamic scenes, following approaches introduced to Gaussians. As an example, 4D Gaussian Splatting \[CVPR 2024\] leverages a Spatial-Temporal Encoder to derive primitive deformations based on the timestep and could similarly be used to extend our work. In the context of applying motion to an existing scene, the explicit nature of our representation would allow us to select and move primitives that make up an object. In theory, the bounded nature of our primitives could aid motion by more distinctly separating individual appearances, and it would be interesting to see the performance in practice. We will discuss it as future work in our revised manuscript.
>
> **Edge-case Failures.** To provide a deeper understanding of which regions and types of objects are well-suited to our approach and which ones are reconstructed better by 3DGS, we created semantic images for ScanNet++ renderings and calculated the per-class PSNR for both approaches. We again consider the first three ScanNet++ scenes, but since coverage is not required as in the depth case, we just use test images. We present the top 3 categories, some select results, and specific failure cases we identified in Table 3\.
>
> Table 3: Comparison of visual errors on specific classes when evaluating on three ScanNet++ scenes.
>
> | *PSNR* | 3DGS | LinPrim | Relative Performance | \#Pixels |
> | :---- | :---- | :---- | :---- | :---- |
> | **Wall (Top 1\)** | 33.74 | **34.76** | \+1.02 | 2.7M |
> | **Floor (Top 2\)** | **27.80** | 25.80 | \-2.00 | 1.1M |
> | **Table (Top 3\)** | 32.65 | **32.88** | \+0.22 | 646k |
> | **Storage Cabinet** | **36.50** | 32.22 | \-4.28 | 283k |
> | **Monitor** | 26.50 | **29.27** | \+2.77 | 223k |
> | **Window** | 23.36 | **23.77** | \+0.41 | 110k |
> | **Computer Tower** | **34.93** | 23.41 | \-11.52 | 15k |
>
> In absolute terms, the failure cases of our approach were the classes of “Storage Cabinet” and “Computer Tower,” in which LinPrim achieved the lowest relative performance compared to 3DGS. On the contrary, the classes that exhibit the strongest view-dependent appearance in the scenes \- windows (+0.41), doors (+1.59), and monitors (+2.77) \- are better reconstructed by LinPrim. There further seem to be types of geometry and appearance LinPrim is better able to capture, e.g., whiteboards (+1.95), blinds (+0.64), and heaters (+0.96).
>
> *Values in brackets describe the relative performance of LinPrim compared to 3DGS as measured by PSNR.*
>
> The results suggest that linear and Gaussian primitives are better able to reconstruct different types of appearance and geometry. Following this, we believe the hybrid primitives proposed by the reviewer could indeed alleviate edge-case failures of either primitive type. Exploring how to bridge the gap between the types of primitive and manage their representations and populations is an interesting avenue for future work. We will add a discussion in the revised paper.
>
> **Anti-Aliasing.** To evaluate the zoom-out capabilities of the proposed anti-aliasing approach, we evaluate the visual quality of single-scale training at multiple smaller scales following Mip-Splatting in Table 4\. While the improvements over the baseline are minor, they are consistent and generalize to zoom-out scenarios. We will add this ablation to our existing discussion on the anti-aliasing approach.
>
> Table 4: Evaluation of zoom-out performance on NeRF synthetic scenes. Results are PSNR values on test set images. Approaches are optimized at full resolution and evaluated on smaller resolutions simulating zoom-outs.
>
> | Chair | Full Res. | 1/2 Res. | 1/4 Res. | 1/8 Res. | Avg. |
> | :---- | :---- | :---- | :---- | :---- | :---- |
> | **No AA** | 35.67 | 34.39 | 30.29 | 27.17 | 31.88 |
> | **LinPrim** | **35.70** | **34.47** | **30.42** | **27.28** | **31.97** |
>
> | Drums | Full Res. | 1/2 Res. | 1/4 Res. | 1/8 Res. | Avg. |
> | :---- | :---- | :---- | :---- | :---- | :---- |
> | **No AA** | 25.96 | 26.42 | 25.34 | 23.14 | 25.22 |
> | **LinPrim** | **25.98** | **26.44** | **25.43** | **23.31** | **25.29** |
>
> | Ficus | Full Res. | 1/2 Res. | 1/4 Res. | 1/8 Res. | Avg. |
> | :---- | :---- | :---- | :---- | :---- | :---- |
> | **No AA** | 34.49 | 34.38 | 31.02 | 26.17 | 31.52 |
> | **LinPrim** | **34.53** | **34.39** | **31.14** | **26.45** | **31.63** |

---

> > ### Comment · Reviewer_Pgum · 2025-08-04
> >
> > Thanks for the detailed response from the authors! All of my concerns have been properly addressed. Please include those additional discussions and evaluations in the revised version of the paper. I'm now pretty positive about this paper.

---

> > > ### Author Response · Authors · 2025-08-04
> > >
> > > Thanks - we'll include them.

---

### Note · Authors · 2025-08-13

We thank the reviewers for the constructive discussion and positive ratings. Reviewers described LinPrim as a compact alternative to Gaussians, with a fully differentiable, analytically grounded rasterizer, comparable fidelity with fewer primitives, and strong potential for future exploration.

*Building on the comments, we expanded our analysis and revised the manuscript accordingly:*

**Bounded and Compact.** Beyond requiring fewer primitives overall, LinPrim’s AABBs are substantially smaller at similar populations, yielding dramatically shorter tile-lists, and far fewer candidates iterated and intersected during rendering, only \~5-10% of the intersections of 3DGS. Despite this compact support, degraded initialization tests show no increased sensitivity, meaning that gradients are reliably propagated.
**Geometry.** LinPrim achieves lower depth errors and correspondingly better geometric reconstructions. TSDF-fused meshes are improved in two out of three scenes.
**View-Dependent Appearance.** Our semantic analysis shows stronger performance on view-dependent/specular classes, and SH ablations indicate less reliance on higher SH bandwidth.
**Compatibility.** LinPrim remains compatible with advances developed for Gaussians. Incorporating MCMC-based densification yields significant quality gains, underscoring that population-control improvements transfer effectively to our linear primitives and allowing them to consistently outperform 3DGS and 3DCS on ScanNet++.
**Anti-Aliasing.** Zoom-out experiments showcase consistent (though modest) performance gains from the introduced filters.

*Apart from the above discussions as well as the corresponding qualitative and quantitative results, the final revision will additionally include:*

**EWA Ray-Space.** Ablation on the performance and quality with and without the ray space approximation.
**Opacity / Density.** Ablation on optimizing densities instead of opacities as primitive features.
**Baselines.** Fully aligned baselines across datasets, full 3DCS results on both datasets, and GS-MCMC on Mip-NeRF 360\.
**LinPrim-MCMC.** Full evaluation of LinPrim-MCMC, as well as qualitative results on both datasets.

The code used for optimization, sampling, and evaluation will be released.

**Takeaway.** Bounded, linear primitives provide compact support and reliable gradients, improve geometry, remain compatible with Gaussian advances, and open clear engineering avenues to close the remaining runtime gap.

---

### Decision · Program_Chairs · 2025-09-17

**Decision:**

Accept (poster)

**Comment:**

This paper introduces LinPrim, a new differentiable representation for volumetric rendering that uses polyhedral primitives, such as octahedra and tetrahedra, as an alternative to the Gaussians in 3DGS.
During the review process, reviewers identified several strengths. They noted the representation is compact, delivering high-quality results with significantly fewer primitives than 3DGS. The new differentiable rasterizer, the solid experimental setup, and the potential for future work were also highlighted as key positives.
However, the reviewers also raised a few concerns about the initial submission:
1. A lack of runtime improvement despite the reduction in primitive count.
2. A limited evaluation, with a need for more comparisons on different scenes and against recent methods (e.g., 3D Convex Splatting).
3. Missing evaluations for anti-aliasing (in zoom-out cases) and for depth/mesh quality.
4. An unclear discussion of the key benefits of the proposed representation over 3DGS.
5. A lack of clarity on the practical memory efficiency.

The authors' detailed rebuttal and the follow-up discussion successfully addressed most of these points, leading all reviewers to be positive about the paper. I agree with the reviewers' final recommendation and recommend this paper for acceptance. I expect the authors will incorporate the new results and discussions from the review period into the final version.